# Phosphorylation of muramyl peptides by NAGK is required for NOD2 activation

Che A. Stafford[1], Alicia-Marie Gassauer[1], Carina C. de Oliveira Mann[1], Maria C. Tanzer[2], Evelyn Fessler[1], Benedikt Wefers[3,4], Dennis Nagl[1], Gunnar Kuut[1], Karolina Sulek[5], Catherine Vasilopoulou[2], Sophia J. Schwojer[1], Andreas Wiest[1], Marie K. Pfautsch[6], Wolfgang Wurst[3,4], Monica Yabal[6], Thomas Fröhlich[1], Matthias Mann[2,5], Nicolas Gisch[7], Lucas T. Jae[1] & Veit Hornung[1,2 ✉]

Bacterial cell wall components provide various unique molecular structures that are detected by pattern recognition receptors (PRRs) of the innate immune system as non-self. Most bacterial species form a cell wall that consists of peptidoglycan (PGN), a polymeric structure comprising alternating amino sugars that form strands cross-linked by short peptides. Muramyl dipeptide (MDP) has been well documented as a minimal immunogenic component of peptidoglycan[1–3]. MDP is sensed by the cytosolic nucleotide-binding oligomerization domain-containing protein 2[4] (NOD2). Upon engagement, it triggers pro-inflammatory gene expression, and this functionality is of critical importance in maintaining a healthy intestinal barrier function[5]. Here, using a forward genetic screen to identify factors required for MDP detection, we identified *N*-acetylglucosamine kinase (NAGK) as being essential for the immunostimulatory activity of MDP. NAGK is broadly expressed in immune cells and has previously been described to contribute to the hexosamine biosynthetic salvage pathway[6]. Mechanistically, NAGK functions upstream of NOD2 by directly phosphorylating the *N*-acetylmuramic acid moiety of MDP at the hydroxyl group of its C6 position, yielding 6-*O*-phospho-MDP. NAGK-phosphorylated MDP—but not unmodified MDP—constitutes an agonist for NOD2. Macrophages from mice deficient in NAGK are completely deficient in MDP sensing. These results reveal a link between amino sugar metabolism and innate immunity to bacterial cell walls.

To identify factors required for MDP recognition, we conducted a forward genetic screen in KBM-7 cells, a near-haploid cell line amenable to gene-trap mutagenesis[7]. To identify cells in which MDP-dependent pro-inflammatory gene expression occurs, we generated a clonal cell line in which mScarlet is linked to the C terminus of endogenous interleukin-1B (IL-1B) (KBM-7-IL-1B^mScarlet), separated by a self-cleaving peptide (Fig. 1a). IL-1B was chosen for its high inducibility following NF-κB activation in these cells. KBM-7-IL-1B^mScarlet cells were treated with L18-MDP, a lipophilic derivative of MDP in which the OH group of the C6 position is esterified with stearic acid[8]. This modification results in enhanced uptake of MDP and hydrolysis of the ester bond within the cell, releasing MDP in the cytoplasm. Cells stimulated with L18-MDP expressed high levels of mScarlet in a NOD2-dependent manner (Fig. 1b,c). Similar results were obtained when stimulating cells with MDP, although much larger amounts of MDP were required to achieve a similar level of activation. As expected, IL-1B–mScarlet expression following treatment with the specific NOD1 agonist iE-DAP[9,10] or stimulation with TNF was not decreased in the absence of NOD2 (Fig. 1c). KBM-7-IL-1B^mScarlet cells were subjected to gene-trap mutagenesis, expanded and stimulated with L18-MDP. We then sorted

these cells according to high or low mScarlet expression. We used deep sequencing to identify gene-trap insertions and calculated enrichment for mutations in genes[11]. This revealed 102 genes ($P_{adj} < 0.05$) that positively regulated MDP-dependent IL-1B–mScarlet expression (Fig. 1d). As well as *IL1B* itself, we identified genes encoding components of the NOD2 pathway[12], including *NOD2*, *RIPK2* and *XIAP*, and many genes encoding factors involved in NF-κB and MAPK signalling (Fig. 1e). In addition to these expected components, we identified *NAGK* as a highly significant hit ($P_{adj} < 3.02 × 10^{-23}$). NAGK is a member of the sugar kinase/Hsp70/actin superfamily, whose members phosphorylate certain substrates in an ATP-dependent manner[13]. In the hexosamine biosynthetic salvage pathway, NAGK mediates the phosphorylation of *N*-acetylglucosamine (GlcNAc) to GlcNAc-6-phosphate, which is then used for biosynthesis of UDP-GlcNAc, the critical component required for *O*-linked-*N*-acetylglucosaminylation and *N*-linked glycosylation[14] (Extended Data Fig. 1a). GlcNAc can be derived from lysosomal degradation of endogenous glycoconjugates or nutritional sources, yet it appears to have only a minor role in UDP-GlcNAc biosynthesis when there are abundant nutritional sources[6,15]. Thus, it seemed unlikely that the loss of MDP-dependent pro-inflammatory gene expression in

[1]Gene Center and Department of Biochemistry, Ludwig-Maximilians-Universität, Munich, Germany. [2]Max-Planck Institute of Biochemistry, Martinsried, Germany. [3]Deutsches Zentrum für Neurodegenerative Erkrankungen e. V. (DZNE), Munich, Germany. [4]Institute of Developmental Genetics, Helmholtz Zentrum München, German Research Center for Environmental Health, Neuherberg, Germany. [5]Novo Nordisk Foundation Center for Protein Research (NNF-CPR), Copenhagen, Denmark. [6]Institute of Molecular Immunology, TUM School of Medicine, Technical University of Munich, Munich, Germany. [7]Bioanalytical Chemistry, Priority Area Infections, Research Center Borstel, Leibniz Lung Center, Borstel, Germany. ✉e-mail: hornung@genzentrum.lmu.de

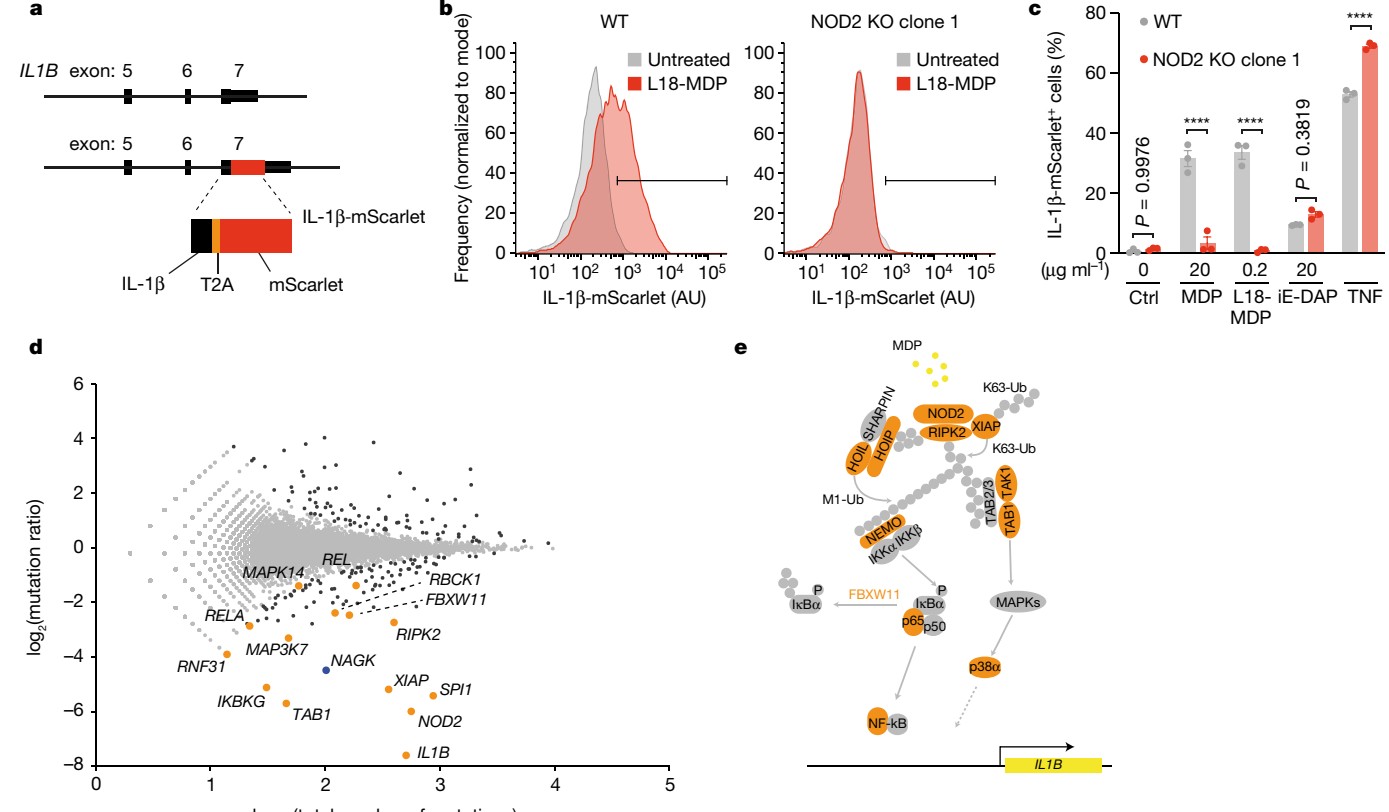

**Fig. 1 | A haploid genetic screen reveals regulators of NOD2. a**, CRISPR engineering of the *IL1B* locus, resulting in an endogenous in-frame fusion of IL-1B and mScarlet separated by a self-cleaving peptide (T2A). **b**, Flow cytometry analysis of IL-1B–mScarlet induction following L18-MDP stimulation of wild-type (WT) and NOD2-knockout (KO) clonal KBM-7-IL-1B^mScarlet cells for 16 h. Data shown for one representative of three independent experiments, indicating gating strategy. AU, arbitrary units. **c**, Flow cytometry analysis of IL-1B–mScarlet induction in wild-type and NOD2-knockout KBM-7-IL-1B^mScarlet cells stimulated as indicated for 16 h. Data are mean ± s.e.m of *n* = 3 independent biological samples. Two-way ANOVA with Šídák's multiple-comparisons test. Ctrl, control. **d**, Genetic screen showing positive

and negative regulators of IL-1B–mScarlet in cells treated with L18-MDP for 16 h. For each gene (dots), the ratio of the mutation frequency in cells with high IL-1B–mScarlet expression versus those with low IL-1B–mScarlet expression is plotted against the combined number of unique mutations identified in both populations of cells. Genes enriched for mutations are shown in black (two-sided Fisher's exact test, false discovery rate (FDR)-corrected *P* value ($P_{adj}$) < 0.05). Genes encoding known components of the NOD2 pathway (orange dots) and the novel regulator NAGK (blue dot) are indicated. **e**, Model of the NOD2 signalling pathway. Products of genes identified in the screen are highlighted in orange; REL and SPI1 are not depicted. ****$P$ < 0.0001.

the absence of NAGK could be attributed to a defect in this pathway. Indeed, no other salvage pathway or UDP-GlcNAc biosynthesis components were identified as significant hits in this screen (Extended Data Fig. 1b). Co-expression analyses revealed that NAGK is co-expressed with genes primarily associated with immune cell functions, such as granulocyte activation (Extended Data Fig. 2). Indeed, across different cell types and tissues, NAGK is expressed mainly within immune cells, most prominently in the myeloid compartment. Together, these data suggested that NAGK has an important role in MDP recognition leading to pro-inflammatory gene expression, independent of its function in the hexosamine salvage pathway.

## NAGK operates upstream of NOD2

To validate the involvement of NAGK in MDP-driven pro-inflammatory signalling, we targeted NAGK in KBM-7 cells using CRISPR–Cas9 and subjected the derived NAGK-deficient cell lines to different stimulations (Fig. 2a,b). NAGK-deficient cells did not respond to the NOD2 stimuli L18-MDP or MDP over a wide range of concentrations. By contrast, NOD1 activation via C12-iE-DAP and other signalling cascades that induced IL-1B–mScarlet expression in KBM-7 cells were unaffected by NAGK deficiency (Fig. 2b). Treatment with IL-1β or TNF, or stimulation

of TLR2–TLR1 (TLR2/1) with Pam₃CSK₄ or PKC with phorbol 12-myristate 13-acetate (PMA) resulted in IL-1B-reporter expression independently of NAGK (Extended Data Fig. 3). The essential and specific requirement for NAGK in the response to MDP was also observed in the production of the chemokines IL-8 and IP-10 (Extended Data Fig. 4a), as well as when measuring a series of pro-inflammatory or antiviral transcripts that are induced following NOD2 and NOD1 stimulation (Extended Data Fig. 4b). Notably, transcript levels of *NOD2* and *RIPK2* were unchanged in the NAGK-knockout cells, suggesting that NAGK had no effect on NOD2 and RIPK2 at a transcriptional stage. Similar results were obtained in NOD2-expressing HEK 293 cells (Fig. 2c). Production of the pro-inflammatory chemokine IL-8 was fully NAGK-dependent when these cells were stimulated with MDP, whereas TNF-dependent activity was independent of NAGK (Fig. 2d). Since these cells provide the opportunity to study NOD2 selectively (no other peptidoglycan (PGN)-sensing pattern recognition receptor (PRR) pathways are functional in these cells), we went on to determine whether the NAGK requirement for NOD2 extended beyond the sensing of the synthetic agonist MDP. To this end, we treated these cells with purified PGN from either *Staphylococcus aureus* or *Escherichia coli*. Genetic ablation of *NAGK* rendered these cells unable to respond to a broad range of PGN concentrations from both bacterial sources, with IL-8 production being

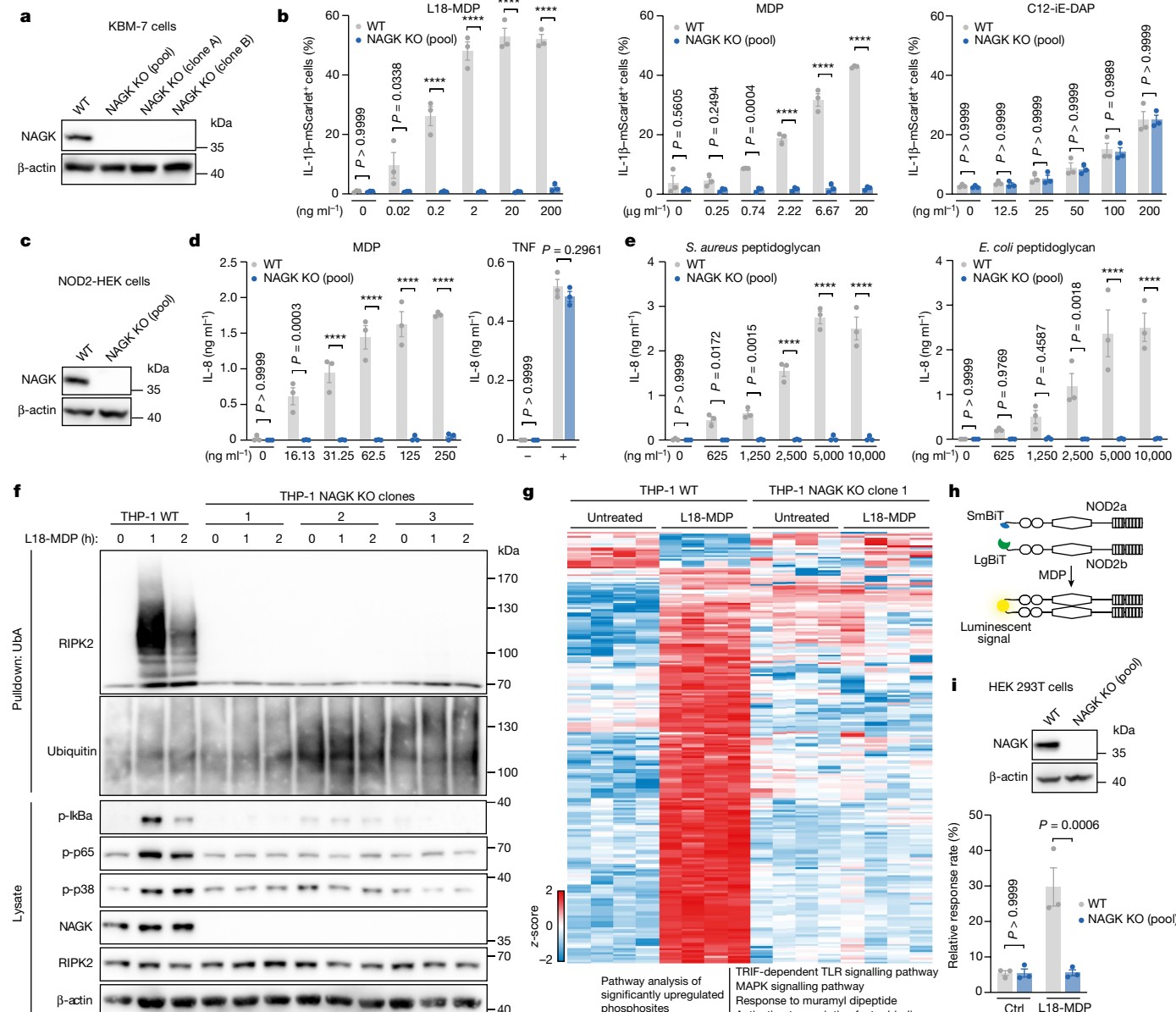

**Fig. 2 | NAGK is required for MDP recognition. a**, NAGK expression in wild-type and NAGK-knockout KBM-7-IL-1B[mScarlet] cells by immunoblot (representative of three independent experiments). **b**, Wild-type and NAGK-knockout KBM-7-IL-1B[mScarlet] cells were treated with the indicated doses of L18-MDP (left), MDP (middle) or C12-ie-DAP (right) for 16 h. Data are mean ± s.e.m of $n = 3$ independent biological samples. Two-way ANOVA with Šídák's multiple-comparisons test. **c**, NAGK expression in wild-type and NAGK-knockout NOD2-expressing HEK cells (representative of two independent experiments). **d**,**e**, Wild-type and NAGK-knockout NOD2-expressing HEK cells were stimulated as indicated and IL-8 production was determined by enzyme-linked immunosorbent assay (ELISA) after 16 h. Data are mean ± s.e.m of $n = 3$ independent biological samples. Two-way ANOVA with Šídák's multiple-comparisons test. **f**, RIPK2 ubiquitination and NF-κB and MAPK activation markers following L18-MDP stimulation in wild-type or NAGK-knockout THP-1 cells. Ubiquitinated proteins were enriched

by pulldown of endogenous ubiquitinated protein bound to an immobilized ubiquitin-associated domain (UbA). Blot represents one of three biological replicates. **g**, Heat map of significantly varying phosphosites in wild-type or NAGK-knockout THP-1 cells treated with L18-MDP for 1 h (Student's *t*-test FDR < 0.05, *z*-scores). Fisher's exact *t*-test of significantly upregulated phosphosites upon L18-MDP treatment ($P < 0.02$; enrichment terms include Gene Ontology Biological Process, KEGG and Gene Ontology Molecular Function). **h**, Schematic representation of the NOD2–NanoBiT luminescence assay. **i**, Immunoblot of NAGK expression in wild-type and NAGK-knockout HEK 239T cells (representative of two independent experiments). Cells were transfected with the constructs shown in **h** and treated with L18-MDP for 24 h. NOD2 dimerization was measured as the luminescence induced by NOD2–NOD2 interaction, normalized with the positive and negative controls. Data are mean ± s.e.m of $n = 3$ independent biological samples. Two-way ANOVA with Šídák's multiple-comparisons test.

completely dependent on the presence of NAGK (Fig. 2e). We obtained analogous results when measuring NF-κB activity using a reporter assay (Extended Data Fig. 5a,b). To study the role of NAGK in a human monocytic cell line, we generated *NAGK*[−/−] THP-1 cells (Extended Data Fig. 5c). THP-1 cells lacking NAGK were unresponsive to MDP stimulation, while their NOD1 response was fully intact (Extended Data

Fig. 5d). Following NOD2 recognition of MDP, the adaptor protein RIPK2 is recruited and ubiquitinated, representing a proximal readout for NOD2 activation upstream of NF-κB[16]. Following stimulation with L18-MDP, THP-1 cells displayed strong ubiquitination of RIPK2, which was absent in the NAGK-knockout cells (Fig. 2f). We also observed a time- and NAGK-dependent increase in the phosphorylation of IκBα,

p65 and p38. Analogous results were obtained in KBM-7 cells that were deficient for NAGK (Extended Data Fig. 5e). To gain a more global view of phosphorylation events following L18-MDP stimulation, we conducted phosphoproteome analysis of wild-type and NAGK-deficient THP-1 cells[17] (Fig. 2g). Following L18-MDP stimulation, we detected 200 phosphosites that were significantly upregulated in wild-type cells and 25 phosphosites that were downregulated. Functional enrichment analysis identified the respective proteins to be involved in innate immune signalling. Conversely, none of these phosphosites were significantly changed in NAGK-deficient THP-1 cells following L18-MDP stimulation. These results are consistent with the notion that NAGK deficiency completely blunts MDP-dependent signal transduction. Finally, to address whether NAGK deficiency directly affected the activation of NOD2, we studied the dimerization or multimerization of NOD2 following stimulation. We generated NOD2 constructs that were N-terminally fused to components of a protein-fragment complementation reporter system, which gains luciferase activity upon complementation (Fig. 2h). Using this system, we observed an increase in luciferase activity in HEK 293T cells expressing these constructs upon L18-MDP treatment. No luminescent signal was observed in cells lacking NAGK (Fig. 2i). Together, these experiments indicated that NAGK is specifically required upstream of NOD2, but not for other pro-inflammatory signalling pathways.

## NAGK phosphorylates MDP

The upstream role of NAGK in relation to NOD2 suggested that NAGK either had a role in MDP uptake or that it directly modified MDP to render it stimulatory. The uptake of a fluorescently labelled muramyl-tripeptide derivative was not affected in the absence of NAGK (Extended Data Fig. 6a,b). However, the direct-modification hypothesis was supported by the notion that GlcNAc—known to be substrate of NAGK[13,18]—is present as a motif in the sugar moiety of MDP (Fig. 3a). In line with this hypothesis, a mutant NAGK(D107V) construct, which is predicted to disrupt enzyme activity[13], did not rescue NOD2 signalling in NAGK-deficient cells (Extended Data Fig. 7). To assess whether NAGK could directly phosphorylate MDP, we expressed recombinant human NAGK in *E. coli* and used it in an in vitro kinase assay with various substrates. The potential transfer of phosphate to the respective substrates was visualized via thin-layer chromatography (TLC). To this end, $[\gamma^{32}P]$ATP was added to the kinase assay, which enabled us to visualize phosphorylated substrates resolved by TLC via phosphor imaging. These assays revealed that NAGK readily phosphorylated MDP and the related molecule *N*-acetylmuramic acid (MurNAc) to similar levels as its bona fide substrate GlcNAc (Fig. 3b). Of note, the TLC assays further revealed that the L-L-isomer of MDP displayed significantly less $\gamma$-phosphate signal in the migrated form (Fig. 3b), indicating that the MDP interaction with NAGK is to some extent stereospecific for the L-D-MDP isomer. This could, in part, explain the finding that the L-L-isomer exerts no stimulatory activity[19] (Fig. 3c). NAGK phosphorylates GlcNAc at the OH group of its C6 position[20]; we thus assumed that MDP is also phosphorylated at this position. Indeed, in line with this notion, L18-MDP, in which the OH group of the C6 position is esterified with stearic acid, was not phosphorylated in this in vitro assay (Fig. 3b). Moreover, 6-amino-MDP, an MDP variant in which the OH group at the C6 position is replaced with an amino group, showed a strongly diminished signal in this assay. This feint band was probably attributable to contamination of the 6-amino-MDP preparation with small amounts of MDP in the chemical synthesis process. Consistent with the notion that a phosphorylated OH group at the C6 position is critically required for the stimulatory activity of MDP, 6-amino-MDP showed no NOD2-stimulatory effect on KBM-7 cells. Only at a high dose was minor activation detected, which we attribute to the contaminating MDP (Fig. 3c). Analysis of MDP that was incubated in the presence of NAGK and ATP by liquid chromatography–tandem mass spectrometry (LC–MS/MS) confirmed that

MDP gained the mass of one phosphoryl group (Fig. 3d). Moreover, NMR analysis of such material revealed that NAGK-mediated phosphorylation of MDP indeed takes place at the hydroxyl group of its C6 position (Extended Data Fig. 8). The fact that L18-MDP cannot be phosphorylated by NAGK in vitro but exerts potent NOD2-stimulatory activity in a NAGK-dependent manner within cells suggests that it is hydrolysed to MDP following its uptake. To study this process, we treated wild-type or NAGK-deficient KBM-7 cells with L18-MDP for 30 min, then lysed the cells and analysed the lysate using targeted LC–MS/MS (Fig. 3e). L18-MDP was detected at similar levels in both wild-type and NAGK-deficient cells (Fig. 3f). Further, the presence of MDP in both cell populations indicated that L18-MDP was being hydrolysed to MDP in the cell. Notably, in line with the idea that NAGK is required to phosphorylate MDP within the cell, we detected a substantial conversion of MDP to 6-phospho-MDP in wild-type cells, whereas there was no such conversion in NAGK-deficient cells. This was underscored by the fact that the NAGK-deficient cells displayed an accumulation of cytosolic MDP compared with wild-type cells (Fig. 3f).

To address whether 6-phospho-MDP was able to stimulate NOD2 in the absence of NAGK, we stimulated cells with either MDP or in vitro-phosphorylated MDP (pMDP). We performed these experiments using digitonin-based membrane permeabilization to account for differences in the net charges and thus membrane permeability of the two molecules. As expected, stimulation with MDP resulted in the activation of wild-type KBM-7 cells, whereas NAGK- or NOD2-deficient cells showed no such activity. However, when we delivered pMDP into these cells, both wild-type and NAGK-knockout cells showed a strong reporter signal, whereas NOD2-knockout cells showed no response (Fig. 3g). We next stimulated unmodified NOD2-expressing HEK cells with increasing doses of these compounds. This revealed that MDP and pMDP displayed high potencies, with slightly higher activity for pMDP (Fig. 3h). The half-maximal effective concentrations ($EC_{50}$) of these molecules were 0.461 nM for MDP and 0.119 nM for pMDP (Fig. 3h). When we stimulated NAGK-deficient cells with these compounds, pMDP retained its potency, with a calculated $EC_{50}$ 0.134 nM. However, the potency of MDP was greatly reduced. Although considerable NF-κB activity was observed at the two highest doses used (20 μM and 4 μM), the calculated $EC_{50}$ value rose by around 4 log to 1,691 nM (Fig. 3h,i). Using IL-8 production as another readout downstream of NOD2 confirmed these observations: wild-type cells responded equally well to digitonin-delivered MDP and pMDP, and a similar response was observed in NAGK-deficient cells only with pMDP stimulation. MDP, however, showed a markedly reduced potency when stimulating NAGK-knockout cells (Extended Data Fig. 9). In summary, these results show that NAGK directly phosphorylates MDP at the OH group of its C6 position to generate phospho-MDP. This phosphorylation event is critically required to render MDP stimulatory for NOD2.

## Mouse BMDMs require NAGK upstream of NOD2

To determine whether NAGK is essential for NOD2 signalling in primary cells, we generated NAGK-deficient mice by targeted CRISPR–Cas9-mediated deletion of exons 2–5 of the *Nagk* gene. *Nagk*$^{-/-}$ mice appeared healthy and of the expected size at 12 weeks of age. To study MDP recognition in these mice, we obtained bone marrow-derived macrophages (BMDMs) from wild-type and *Nagk*$^{-/-}$ mice, which we studied under IFNγ-primed conditions. Consistent with the results obtained with cell lines, *Nagk*$^{-/-}$ BMDMs did not produce TNF or IL-6 upon MDP stimulation, as measured by ELISA (Fig. 4a). NAGK deficiency specifically affected MDP recognition, since BMDMs from both wild-type and *Nagk*$^{-/-}$ mice were able to produce TNF and IL-6 to similar levels when stimulated with the TLR2/1 ligand Pam$_3$CSK$_4$. We next measured the production of a number of pro-inflammatory or antiviral transcripts upon NOD2,

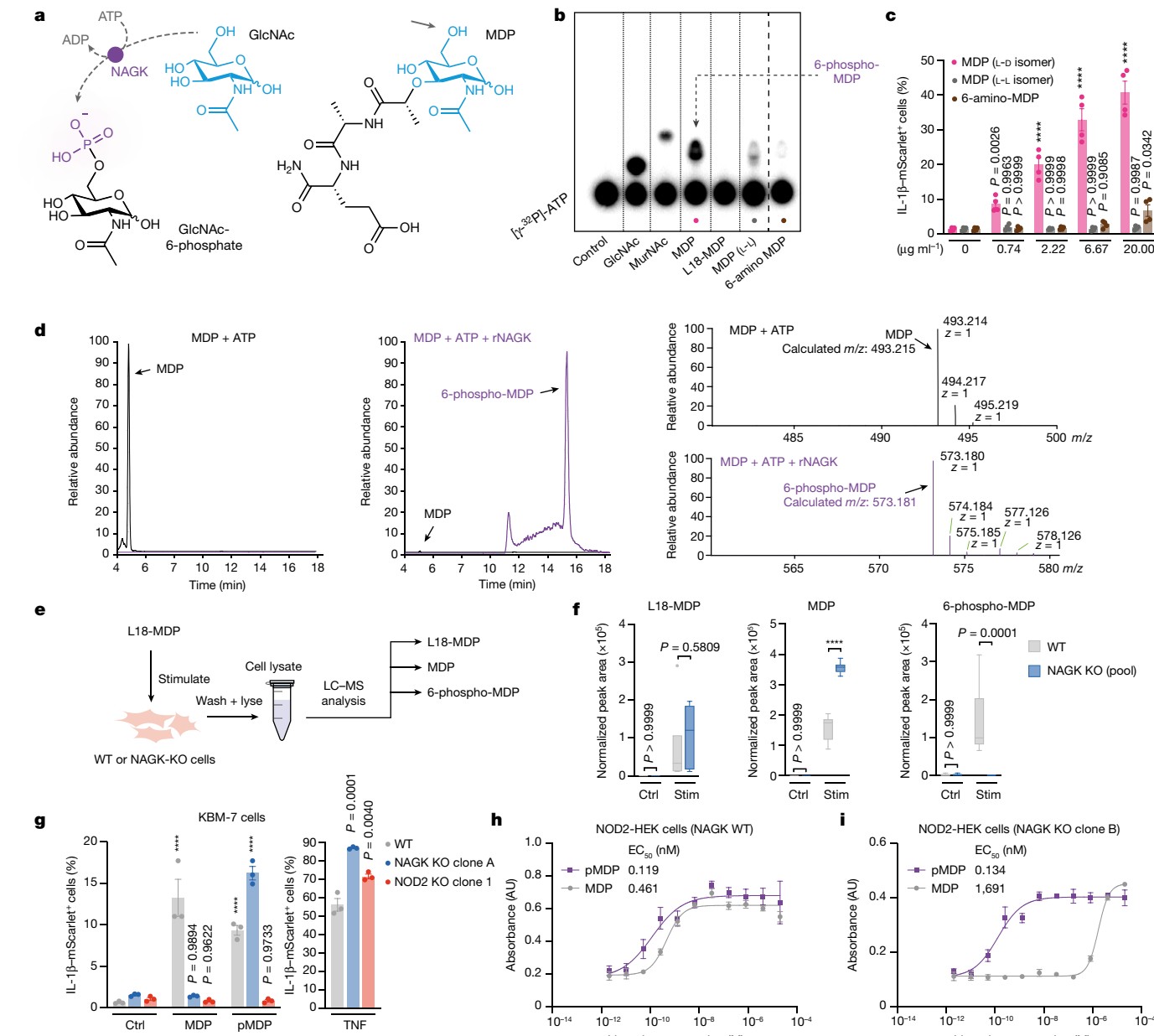

**Fig. 3 | NAGK phosphorylates MDP to render it NOD2-agonistic. a**, Left, GlcNAc conversion to GlcNAc-6-phosphate in a NAGK-dependent manner. Right, MDP. Sugar moieties are indicated in blue. The arrow indicates the phosphorylatable OH group in the C6 position of MDP. **b**, In vitro kinase reactions with indicated ligands were analysed by TLC. The dashed line indicates where the image has been cropped to remove superfluous lanes. The image represents three biological replicates. **c**, Wild-type KBM-7-IL-1B^mScarlet cells were treated with indicated doses MDP (L-D isomer), MDP (L-L isomer) or 6-amino MDP for 16 h. Data are mean ± s.e.m of $n = 4$ independent biological samples. Two-way ANOVA with Dunnett's multiple-comparisons test. Comparisons were made with untreated controls, which are depicted three times for comparability. **d**, Extracted ion chromatogram of the MDP in vitro kinase reaction without (left) or with (middle) recombinant NAGK (rNAGK). Chromatographic peaks indicate either MDP (black) or 6-phospho-MDP (purple). Right, corresponding mass spectra from these peaks, with theoretical

and detected mass. **e**, Schematic of the cellular L18-MDP, MDP and 6-phospho-MDP content analysis by LC–MS. **f**, L18-MDP, MDP and 6-phospho-MDP analysis in wild-type and NAGK-knockout KBM-7 cells treated with L18-MDP. Tukey-style box plots of $n = 6$ independent biological samples. Two-way ANOVA with Šídák's multiple-comparisons test. Stim, stimulated. **g**, IL-1B–mScarlet expression in wild-type, NAGK-knockout or NOD2-knockout KBM-7-IL-1B^mScarlet cells that were untreated or stimulated with MDP, pMDP or TNF in digitonin buffer for 16 h. Data are mean ± s.e.m of $n = 3$ independent biological samples. Two-way ANOVA (MDP and pMDP) or one-way ANOVA (TNF) with Dunnett's multiple-comparisons test. Comparisons to untreated controls (MDP or pMDP) or wild-type cells (TNF). **h,i**, NF-kB reporter activity in wild-type or NAGK-deficient NOD2-expressing HEK cells stimulated with MDP or pMDP in digitonin buffer as indicated for 16 h. Data are mean ± s.e.m of $n = 3$ independent biological samples. A four-parameter dose-response curve was fitted to calculate half-maximal effective concentration ($EC_{50}$).

NOD1 or TLR2/1 stimulation (Fig. 4b). Wild-type BMDMs exhibited strong induction of these transcripts, whereas $Nagk^{-/-}$ cells were transcriptionally inactive upon MDP treatment. The wild-type and $Nagk^{-/-}$ populations showed similar signalling responses to NOD1 and TLR2/1 stimulation.

Using ubiquitin pulldowns to determine RIPK2 ubiquitination as a proximal readout for NOD2 activation in both wild-type and $Nagk^{-/-}$ cells, we observed strong RIPK2 ubiquitination in wild-type BMDMs, but no RIPK2 ubiquitination in BMDMs deficient for NAGK. This lack of RIPK2 ubiquitination in the $Nagk^{-/-}$ cells was paralleled by

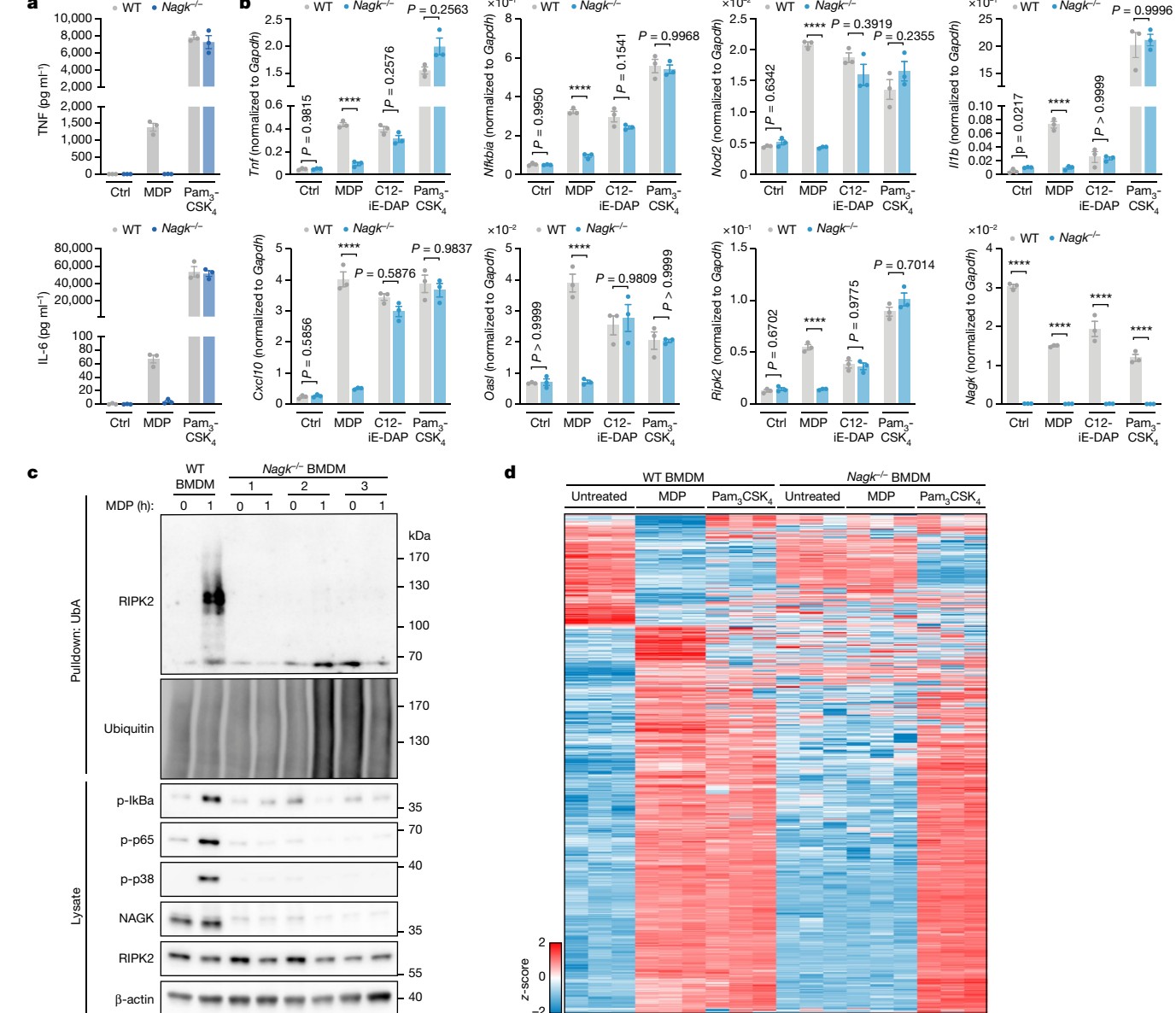

**Fig. 4 | NAGK is essential for NOD2 signalling in mouse BMDMs. a**, ELISA for TNF and IL-6 production in wild-type and *Nagk*⁻/⁻ mouse BMDMs treated stimulated as indicated for 24 h. Data are mean ± s.e.m of *n* = 3 independent biological samples. **b**, Quantitative PCR analysis of the indicated transcripts in wild-type and *Nagk*⁻/⁻ BMDMs treated with either MDP, C12-iE-DAP or Pam₃CSK₄ for 5 h. Gene expression levels were normalized to *Gapdh*. Data are mean ± s.e.m of *n* = 3 independent biological samples. Two-way ANOVA on log-transformed data with Šídák's multiple-comparisons test. **c**, RIPK2 ubiquitination and NF-κB and MAPK activation markers following MDP

stimulation of wild-type and *Nagk*⁻/⁻ BMDMs. Ubiquitinated proteins were enriched by pulldown of endogenous ubiquitinated protein bound to immobilized UbA. Blot represents one of three biological replicates for the wild type and presents all three *Nagk*⁻/⁻ mice. **d**, Heat map of significantly varying phosphosites in wild-type BMDMs treated with MDP for 1 h (Student's *t*-test FDR < 0.01, *z*-scores). *Z*-scores for these phosphosites in *Nagk*⁻/⁻ BMDMs treated with either MDP in both genotypes treated with Pam₃CSK₄ for 1 h were included. All BMDM experiments were conducted under IFNγ-primed conditions from three mice per condition.

the absence of IκBα, p65 and p38 phosphorylation, compared with a strong induction of phosphorylation in the wild-type MDP-treated samples (Fig. 4c). Next, we performed phosphoproteomic analysis of wild-type and *Nagk*⁻/⁻ BMDMs treated with either MDP or Pam₃CSK₄ as a control (Fig. 4d). Following MDP stimulation, 1,041 phosphosites were detected to be significantly upregulated and 298 were downregulated in wild-type cells. By contrast, only four phosphosites were significantly changed in the *Nagk*⁻/⁻ BMDMs following MDP stimulation. As expected, Pam₃CSK₄ treatment resulted in a robust shift in the phosphoproteome of both wild-type and *Nagk*⁻/⁻ BMDMs. Together, these results indicate that primary mouse macrophages require NAGK to sense NOD2-agonistic stimuli.

## Conclusion

Here we show that the amino sugar kinase NAGK phosphorylates MDP at the hydroxyl group of the C6 position of its MurNAc moiety. This NAGK-dependent phosphorylation of MDP is critically required to render this pathogen-associated molecular pattern visible to the host's innate immune system. In all cells in which NOD2 could be engaged—including primary macrophages—NAGK deficiency completely blunted the response to MDP. Complex peptidoglycan preparations from both Gram-negative and Gram-positive bacteria were also dependent on NAGK for their NOD2 response. The requirement for NAGK is derived solely from its ability to directly phosphorylate MDP, since

in vitro-phosphorylated MDP could bypass the requirement for cellular NAGK. We hypothesize that MDP phosphorylation is a critical prerequisite for it being a direct ligand for NOD2. However, in the absence of biochemical data demonstrating binding, this remains speculative, and it is conceivable that other intermediate steps are required for NOD2 to recognize phosphorylated muramyl peptides.

Many pathogenic bacteria have evolved to specifically modify the 6′-OH group of the MurNAc moiety utilizing PGN-specific *O*-acetyltransferases. This modification strongly affects their pathogenicity in that it increases the resistance of their PGN to lysozyme[21]. *S. aureus*, for example, critically depends on the expression of its *O*-acetyltransferase A (OatA) for its pathogenicity. Further, *S. aureus* deficient in OatA elicits a far stronger pro-inflammatory cytokine profile in infected hosts[22]. Although this can in part be attributed to increased ligand availability due to lysozyme sensitivity, our results suggest that PGN devoid of *O*-acetylated MurNAc would also gain immunostimulatory activity owing to it being a substrate for NAGK phosphorylation.

The functional connection of NAGK to the recognition of bacterial cell wall components is well aligned with its primordial function in prokaryotes. In bacteria, the NAGK orthologue has an important role in the recycling of PGN components. This process is vital to allow the remodelling of the bacterial cell wall during growth and adaptation[23]. NAGK has been retained throughout evolution, even in organisms that do not harbour GlcNAc-containing bacterial cell walls or chitin exoskeletons. Conversely, NOD2 first appeared in vertebrates (Extended Data Fig. 10a). Since MDP is readily converted to 6-phospho-MDP within the cell—and given the primordial presence of NAGK and its expression in immune cells—we speculate that NOD2 evolved to sense 6-phospho-MDP rather than MDP. In line with this notion, NAGK-deficient cells could be complemented with NAGK derived from *Drosophila* to restore their NOD2 response (Extended Data Fig. 10b). In this regard, it will be interesting to explore whether additional evolutionary remnants of PGN catabolic enzymes exist in vertebrates and whether these factors also contribute to self versus non-self recognition.

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

# Methods

## Cell culture

KBM-7 cells were cultured in Iscove's Modified Dulbecco's Medium (IMDM) (Thermo Fisher Scientific) supplemented with 10% heat-inactivated fetal calf serum (FCS) (Thermo Fisher Scientific) and 1% penicillin-streptomycin–glutamine solution (Thermo Fisher Scientific). THP-1 cells were cultured in RPMI medium 1640 containing the same supplements. HEK 293T (293T) were maintained in Dulbecco's modified Eagle's medium (DMEM) (Thermo Fisher Scientific) containing the same supplements. NOD2-HEK cells expressing human NOD2 and an NF-κB-SEAP reporter cassette (HEK-Blue-NOD2 cells) were obtained from Invivogen. Cells were cultured in DMEM supplemented with 10% heat-inactivated FCS, zeocin, blasticidin and normocin, as suggested by Invivogen. Cell lines were initially tested negative for mycoplasma contamination but were not tested routinely thereafter.

## Generation of *Nagk*[−/−] mice

*Nagk*[−/−] mice were generated by CRISPR–Cas9-mediated gene editing in zygotes as described[24]. In brief, pronuclear stage zygotes were obtained by mating C57BL/6J males with superovulated C57BL/6J females. Embryos were then electroporated using the NEPA21 electroporator and a 1 mm electrode with a *Nagk*-specific CRISPR–Cas9 ribonucleoprotein (RNP) solution consisting of 200 ng μl⁻¹ SpCas9 protein (Integrated DNA Technologies (IDT)), 3.0 μM of each crRNA (IDT), and 6.0 μM tracrRNA (IDT). Protospacer sequences were GAGTGTAGCAGGCACTAAAC in intron 1–2, and ACGATTGAAACTGAAGGTTA in intron 5–6. After electroporation, zygotes were transferred into pseudopregnant CD-1 foster mice. To exclude additional unwanted modifications, putative off-target sites of the *Nagk*-specific crRNAs were predicted using the CRISPOR online tool[25]. Genomic DNA from $F_1$ mice was PCR-amplified and verified by Sanger sequencing and did not show additional sequence variation.

All mice were handled according to institutional guidelines approved by the animal welfare and use committee of the government of Upper Bavaria and housed in standard cages in a specific pathogen-free facility ($21 \pm 1$ °C, on a 12-h light/dark cycle, with average humidity of around 55%) with ad libitum access to food and water in the animal facility at the Centre for Neuropathology.

## Generation and culture of BMDMs

BMDMs were generated from the femur and tibiae of wild-type and *Nagk*[−/−] C57BL/6 mice aged 12 weeks and cultured for 6 days in DMEM supplemented with 8% FCS, 20% L929 supernatant, and 1% penicillin-streptomycin–glutamine solution. After 6 days, cells were detached by cell scraping and replated into 10-cm dishes for phospho-proteomics and ubiquitin pulldowns ($10^7$ cells per dish), 6-well plates for quantitative PCR (qPCR) ($10^6$ cells per well), and 24-well plates for ELISA ($2.5 \times 10^5$ cells per well).

## BMDM stimulation protocol

BMDMs were primed with murine IFNγ (5 ng ml⁻¹, ImmunoTools) for 16 h, followed by the medium being replaced with fresh IFNγ (5 ng ml⁻¹) for 2 h before treatment with indicated stimulations: MDP (10 μg ml⁻¹, Invivogen), C12-iE-DAP (400 ng ml⁻¹, Invivogen), Pam₃CSK₄ (500 ng ml⁻¹, Invivogen). Stimulation for ELISA (24 h), phosphoproteomics (1 h), qPCR (5 h) and ubiquitin pulldowns (1 h).

## Generation of KBM-7 IL-1B-mScarlet reporter cells

Near-haploid KBM-7 cells were electroporated with four plasmids to generate KBM-7-IL-1B[mScarlet] reporter cells. A pRZ-CMV-Cas9 plasmid encoding for Cas9, pBlueScript II SK (+) containing a donor cassette consisting of a T2A-threonine-glycine-mScarlet and two pL-U6-guide RNA (gRNA) plasmids encoding single guide RNAs (sgRNAs) against IL-1B. The donor cassette was flanked by the sgRNA sites and homology arms mapping to around 400 bp upstream and 500 bp downstream of the stop codon at the *IL1B* locus. The sgRNA recognition sites cut upstream and downstream of the *IL1B* stop codon and release the knock-in cassette after uptake into the transfected cells. The sgRNA recognition sequences near the *IL1B* stop codon were eliminated in the donor DNA sequence via synonymous mutations. Single-cell clones were then selected for activity following different pathogen-associated molecular pattern stimulations.

## Haploid genetic screening for identification of NOD2 regulators

Ultra-deep genome-wide mutagenesis of haploid KBM-7 cells was carried out as described previously[26]. In brief, a BFP-containing variant of gene-trap retrovirus was transfected and produced in 293T cells and harvested initially after 48 h, followed by six additional collections. Retroviral particles were concentrated by ultracentrifugation at 88,880$g$ for 2 h at 4 °C and stored at 4 °C overnight. To generate random genomic mutations by the gene-trap virus, near-haploid KBM-7-IL-1B[mScarlet] reporter cells were seeded at $2.5 \times 10^6$ cells per ml and repeatedly transduced by spinfection (1,200$g$, 90 min, 21 °C). The resulting library of mutants was used for genetic screens.

To identify NOD2 regulators, mutagenised KBM-7-IL-1B[mScarlet] reporter cells were seeded at $2.5 \times 10^6$ cells per ml and treated with L18-MDP (200 ng ml⁻¹) for 16 h. Cells were collected and washed with PBS. Cells were passed through a 40-μm cell strainer (Greiner, 542040) before fixation with one pellet volume methanol-free 4% formaldehyde solution (Thermo Scientific) for 13 min at 37 °C. Fixation was stopped with PBS containing 1% FCS, and cells were once more passed through a 40-μm cell strainer and counted. Cells were resuspended to a final concentration of $10^8$ cells per ml, and the DNA was stained using DAPI (Sigma-Aldrich, D9542) at a final concentration of 5 μg ml⁻¹. After three washes with PBS containing 1% FCS, cells were resuspended at $10^8$ cells per ml in PBS and 1% FCS, stored at 4 °C, and sorted on a BD Fusion cell sorter (BD Biosciences) using a 70-μm nozzle. Haploid cells were identified based on DNA content in the DAPI channel, and of those, approximately $10^7$ cells of the bottom 4% IL-1B–mScarlet[low] and top 4% IL-1B–mScarlet[high] cells were sorted for isolation of genomic DNA. Gating strategies for the flow cytometry-based studies are provided in Supplementary Fig. 2.

## Analysis of the haploid genetic screen

For the genome-wide haploid genetic screen, mutagenised KBM-7-IL-1B[mScarlet] cells were interrogated phenotypically by mScarlet fluorescence prior to the identification of gene-trap mutations by deep sequencing. Alignment to the human genome (hg19) was carried out as reported[11] and yielded a total of 1,640,694 unique mutations (averaging approximately 101 mutations per gene) in the sense orientation of genes in cells with either elevated or diminished mScarlet signal. For each affected gene, a two-sided Fisher's exact test was employed to calculate enrichment for mutations in the populations with either elevated or diminished mScarlet fluorescence. Resulting $P$-values were adjusted for multiple testing based on the Benjamini–Hochberg method. For each gene, the fraction of unique sense insertions mapping to that gene identified in the mScarlet-high cells was divided by the corresponding fraction of unique sense insertions in the mScarlet-low cells. For data visualization, per gene, this mutation ratio ($y$-axis) was subsequently plotted against the combined number of unique mutations identified in the gene in the mScarlet-high and mScarlet-low cells ($x$-axis).

## Cell stimulation assays

Unless otherwise indicated, the following concentrations for cell stimulation assays (readouts: flow cytometry, ELISA or qPCR) were used: MDP (20 μg ml⁻¹, Invivogen), L18-MDP (200 ng ml⁻¹, Invivogen), iE-DAP (20 μg ml⁻¹, Invivogen), C12-ie-DAP (200 ng ml⁻¹), TNF (6.25 ng ml⁻¹, PeproTech). KBM-7 cells ($2.5 \times 10^6$ per ml) (100 μl per well final volume) were stimulated for 16 h at 37 °C. For flow cytometry studies,

mScarlet-positive cells were analysed on a BD LSR Fortessa. For ELISA experiments, cells were spun, and the supernatant was collected. THP-1 cells ($2.5 \times 10^6$ per ml) (100 µl per well final volume) were stimulated for 16 h at 37 °C. For ELISA experiments, cells were spun, and the supernatant was collected.

One-hundred thousand NOD2-HEK cells per ml (50 µl per well) were seeded in clear 96-well plates. After 48 h, cells were changed into HEK-Blue detection medium (Invivogen) containing the indicated stimuli. Peptidoglycan (PGN) from *S. aureus* was digested with lysozyme (20 µg ml$^{-1}$) in 50 mM Tris (pH 6.8), incubated at 37 °C for 45 min followed by heat inactivation of the lysozyme at 95 °C for 5 min and clarified by centrifugation. Cleared supernatant containing digested PGN was then used for treatment. Ultrapure, soluble PGN from *E. coli* was used directly to stimulate the cells. After 16 h of stimulation, absorbance at 620 nm was measured using a plate reader (Tecan Spark 20M). For digitonin permeabilization experiments, cells were treated with permeabilization buffer as previously described[27]: 50 mM HEPES (pH 7.0), 100 mM KCl, 3 mM MgCl$_2$, 0.1 mM DTT, 85 mM sucrose, 0.2% BSA, 1 mM ATP and 0.1 mM GTP with or without 10 µg ml$^{-1}$ digitonin (Sigma) supplemented with indicated ligands for 10 mins. Buffer was then removed, and cells were placed in medium for 16 h.

## Analytical flow cytometry
Cells were analysed on a BD LSR Fortessa. For data collection the BD FACSDiva 8.0.1 and 8.0.2 software was used. Flow cytometry data were analysed using FlowJo 10.7. All histograms are shown normalized to mode. The percentage of IL-1β-mScarlet$^+$ cells was derived from gating on both wild-type and NAGK-deficient untreated cell populations.

## UbA pulldown
Ten million BMDMs or twenty million THP-1 or KBM-7 cells were stimulated with MDP (for BMDMs, 10 µg ml$^{-1}$) or L18-MDP (for THP-1s and KBM-7s, 200 ng ml$^{-1}$) for the indicated times, washed in PBS, and lysed in 500 µl DISC buffer (150 mM NaCl, 50 mM Tris pH 7.5, 10% glycerol, 1% Triton X-100) with cOmplete protease inhibitor cocktail (Roche) and 10 mM *N*-ethylmaleimide for 20 min on ice. Samples were clarified by centrifugation at 17,000*g* for 10 min and added directly to 20 µl packed glutathione Sepharose beads pre-bound with 100 µg GST–UbA. Beads were incubated on a rotating wheel at 4 °C overnight, washed 3 times with DISC buffer, and eluted with 2 × SDS sample buffer (4% SDS, 20% glycerol, 120 mM Tris-Cl pH 6.8, 0.02% bromophenol blue).

## CRISPR–Cas9-mediated knockout cell line generation
THP-1, KBM-7, NOD2-HEK and HEK 293T cells were gene-targeted with RNPs for either NAGK or NOD2. The following gRNAs were used: NAGK gRNA1: 5′-GCCTAGGGCCTATCTCTGAG-3′, NAGK gRNA2: 5′-TTAATCACCACCGATGCCGC-3′, NOD2 gRNA1: 5′-GGACTGGCT GCTGTCCTGGG-3′, NOD2 gRNA2: 5′-CGAGCACATTTCACAACCTG-3′. CRISPR–Cas9–RNPs were assembled by annealing synthetic, chemically stabilized crRNA:tracrRNA pairs (IDT) at 95 °C for 5 min and incubation at room temperature for 30 min. gRNAs were then mixed with recombinant NLS–Cas9 protein for 10 min at room temperature. For every gRNA pair, 40 pmol Cas9 was added for each 100 pmol of gRNA. For nucleofection, 1 million cells were resuspended in 20 µl of nucleofection buffer P3 (Lonza) for KBM-7 cells or buffer SG for THP-1 cells, NOD2-HEK cells, and HEK 293T. This buffer was supplemented with RNPs and nucleofected with the following programs: THP-1 (program FF-100), KBM-7 (program EH-100), NOD2-HEK cells (CM-130), HEK 293T (CM-130). After nucleofection, cells were collected from the nucleofection cuvettes with warm medium and transferred to a 6-well plate for KBM-7 and THP-1 and a 24-well plate for NOD2-HEK cells and HEK 293T cells. Cells were rested for 48 h and subjected to minimal dilution cloning. When colonies became visible after approximately three weeks, clones were collected and either analysed by western blotting or MiSeq[28]. The parental cell lines were used as wild-type controls. Unless otherwise indicated, pool knockouts were used in experiments.

## Generation of doxycycline-inducible cell lines
Human NAGK or human NOD2 was amplified from cDNA derived from KBM-7 cell lysate and cloned into a doxycycline-inducible (dox-on) lentiviral plasmid using conventional Gibson cloning (pLI_hsNAGK_ WT_Puro). HA-tagged *Drosophila* NAGK (CG6218) was codon-optimized and synthesized by IDT, then cloned into dox-on lentiviral plasmid using conventional Gibson cloning (pLI_dmNAGK_Puro). The NAGK sugar-binding mutant (D107V) (pLI_hsNAGK_D107V_Puro) was cloned from the wild-type sequence and mutated using Gibson mutational cloning. The indicated genotypes were transduced, and plasmid integration was selected for using puromycin (1 µg ml$^{-1}$). The polyclonal cell population was then used for further experiments. For stimulation, cells were treated with doxycycline (1 µg ml$^{-1}$) for 5 h preceding stimulation.

## Radiolabelled NAGK kinase activity assay
The ability of human NAGK protein to phosphorylate various amino sugars was analysed as previously described for the bacterial MurK protein[29]. In brief, 10 nM NAGK were incubated with 0.1 mM or 1 mM substrate, 0.5 mM ATP and trace amounts of [$_\gamma$$^{32}$-P]ATP (Hartman Analytic) in reaction buffer (50 mM HEPES pH 7.5, 50 mM NaCl, 10 mM MgCl$_2$). Reactions were incubated for 1 h at 37 °C, and 1 µl reaction products were spotted on TLC (silica gel 60 F254, Merck). Butanol-methanol-ammonia-water (5:4:2:1 [vol/vol/vol/vol]) was used as running buffer, and TLC plates were analysed by phosphor imaging (Typhoon FLA 9000, GE Healthcare).

## In vitro MDP chromatography and mass spectrometry
For MDP, pMDP and NAGK identification samples were digested using trypsin before LC–MS/MS analysis. Proteins were reduced using 5 mM dithioerythritol at 37 °C for 30 min, and cysteines were alkylated with iodoacetamide (final concentration 15 mM) for 30 min in the dark. Digestion was performed using 7.4 ng sequencing grade modified porcine trypsin (Promega). For MDP and pMDP detection an EASY-nLC 1200 system (Thermo Fisher Scientific) coupled to an Orbitrap Elite mass spectrometer (Thermo Fisher Scientific) was used. Samples were separated at 300 nl min$^{-1}$ on an EASY-Spray column (PepMap RSLC C18, 15 cm × 75 µm internal diameter, Thermo Fisher Scientific). Solvent A consisted of 2% acetonitrile in 0.1% formic acid and solvent B of 95% acetonitrile in 0.1% formic acid. For separation, 2 consecutive gradients from 2% to 35% solvent B in 30 min and from 35% to 80% B in 5 min were applied. Mass spectra acquisition was done in positive ion mode using a scan range from 250 to 1,800 *m/z* at a resolution of 120k. Extracted ion chromatograms were generated using Xcalibur Qual Browser 4.1.31.9 (Thermo Fisher Scientific) using *m/z* 493.215 for MDP and *m/z* 573.181 for phospho-MDP at mass tolerances of 10 ppm. For NAGK identification, an Ultimate 3000 nano-LC system (Thermo Fisher Scientific) coupled to a QExactive HF-X mass spectrometer (Thermo Fisher Scientific) was used. Samples were transferred to a trap column (Acclaim PepMap 100, 100 µm × 2 cm, nanoViper C18, Thermo Fisher Scientific) at a flow rate of 20 µl min$^{-1}$ and separated at 250 nl min$^{-1}$ with an EASY-Spray column (PepMap RSLC C18, 50 cm × 75 µm internal diameter, Thermo Fisher Scientific). Solvent A consisted of 0.1% formic acid and solvent B of 0.1% formic acid in acetonitrile. For separation, two consecutive gradients from 3% to 25% solvent B in 30 min and from 25% to 40% B in 5 min were applied. Mass spectra acquisition was performed in positive ion mode using a scan range from *m/z* 350 to 1,600 at a resolution of 60k. Data-dependent HCD MS/MS spectra were collected at 15k resolution. Spectra were analysed using MASCOT V2.6.1 (Matrix Science) and the human subset of the UniProt database. To further check for potential *E. coli* impurities in the recombinant NAGK preparation, the spectra were further analysed using the UniProt *E. coli* sequence database.

## Cellular L18-MDP, MDP and 6-phospho-MDP extraction protocol

Five million KBM-7 wild-type and NAGK pool knockout cells were collected either untreated or stimulated with L18-MDP (200 ng ml$^{-1}$) for 30 min, then washed with 37 °C PBS twice, then snap frozen in liquid nitrogen and stored at −80 °C. Cells were extracted using 300 µl of cold 90% methanol (LC–MS grade Optima, Fisher Scientific). Labelled D5-tryptophan (CDN Isotopes) was added to the samples (10 µl of 0.005 mg ml$^{-1}$) as internal standard. After brief vortexing, samples were sonicated in a Bioruptor (BioNordika), that was set to 4 °C for 15 cycles of 30 s each, with 30 s break, followed by 15 min shaking at 2,000 rpm at 4 °C in an Eppendorf ThermoMixer (Fisher Scientific). After centrifugation at 15,000 g for 15 min at 4 °C (Eppendorf Microcentrifuge, Fisher Scientific) the supernatant was collected to a new, pre-chilled 1.5 ml tube and dried at room temperature (SpeedVac Vacuum Concentrator, Thermo Fisher Scientific). Dried samples were reconstituted in 50 µl H$_2$O (LC–MS grade Optima, Fisher Scientific), vortexed and shaken at 2,000 rpm for 15 min at 4 °C. After centrifugation at 15,000 g for 15 min at 4 °C, the supernatant was collected in LC–MS vials with glass inserts (Phenomenex) and stored at −80 °C until LC–MS analysis.

## Cellular L18-MDP, MDP and 6-phospho-MDP LC–MS analysis

Samples were thawed in the fridge and kept at 4 °C in the autosampler during the LC–MS acquisition. Acquisition was performed on an LC Vanquish system coupled to an Orbitrap 480 Exploris MS (Thermo Fisher Scientific) equipped with a heated electrospray ionization source (HESI). Ten microlitres of the cell extract was injected and molecules were separated on a Luna Omega Polar C18, 100 × 2.1 mm column with particle size of 3 µm and pore size of 100 Å with porous polar C18 guard cartridge (Phenomenex) in a 15-min gradient (buffer A: 0.1% formic acid in H$_2$O, buffer B: 0.1% formic acid in acetonitrile; flow rate at 300 µl min$^{-1}$; 0% B in the first 0.5 min, 0–99% B in 9.5 min, 99% B maintained for 2 min, lowered to 0% in 1 min and equilibrated with 0% B for 2 min). Column temperature was maintained at 45 °C. Profile data was acquired in negative mode with the following settings: spray voltage 3.4 kV, heated capillary temperature at 350 °C, sheath gas 60 arbitrary units, aux gas 20 arbitrary units, sweep gas 0, funnel RF level at 60%. Full-MS spectra (50–1,000 m/z) were acquired after accumulation of 3 × 10$^6$ ions in the Orbitrap (AGC target 300%, maximum injection time of 100 ms, microscans 1) at 120,000 resolution.

Standard compounds of interest were processed alongside the samples and injected for mass to charge and retention time elucidation; D5-tryptophan (208.10–208.20; 3.0–4.0 min), L18-MDP (757.40–757.50; 9.0–10.0 min), MDP (491.20–491.21; 2.0–3.0 min) and pMDP (571.10–571.20; 2.0–2.9 min). Blanks with internal standard were used as negative test in case of chromatographic impurity.

Peak height (AH) and peak area (AA) were extracted from MS raw files for targeted analysis for D5-tryptophan, L18-MDP, MDP, and pMDP with mass to charge and retention time specified above using Xcalibur Processing Setup Quan (version 4.2.47, Thermo Scientific). Smoothing point 1, baseline window 50, area noise factor 5, peak noise factor 10, highest peak with minimum peak height 3.0 were used. L18-MDP, MDP, and 6-phospho-MDP relative abundance was normalized to the level of the internal standard D5-tryptophan in the respective sample. The abundance of each molecule is represented as normalized peak area (AA).

## Phosphoproteomics

We applied the EasyPhos protocol[17] to enrich for phosphopeptides. For THP-1 samples: 8 × 10$^6$ THP-1 wild-type or NAGK-KO clone 1 cells were stimulated in 4 replicates with L18-MDP (200 ng ml$^{-1}$) for 1 h, then washed with ice-cold TBS and lysed in 2% sodium deoxycholate and 100 mM Tris-HCl pH 8.5 and boiled immediately. For BMDMs, three wild-type and *Nagk*$^{-/-}$ BMDM cell populations were either left untreated or treated with MDP (10 µg ml$^{-1}$) or Pam$_3$CSK$_4$ (500 ng ml$^{-1}$) for 1 h then

processed as above. After sonication, protein amounts were adjusted to 1 mg before reduction (10 mM tris(2-carboxy(ethyl)phosphine) (TCEP)), alkylation (40 mM 2-chloroacetamide) and digestion with trypsin and lysC (1:100, enzyme:protein, w:w) overnight. Isopropanol (final concentration 50 %), trifluoroacetic acid (TFA, final concentration 6%), and monopotassium phosphate (KH$_2$PO$_4$, final concentration 1 mM) were added to the rest of the digested lysate. Lysates were shaken, then spun for 3 min at 2,000g, and supernatants were incubated with TiO$_2$ beads for 5 min at 40 °C (1:10, protein:beads, w:w). Beads were washed 5 times with isopropanol, and 5% TFA and phosphosites were eluted off the beads with 40% acetonitrile (ACN) and 15% of ammonium hydroxide (25% NH$_4$OH) on C8 stage tips. After 20 min of SpeedVac at 45 °C, phosphosites were desalted on SDB-RPS stage tips and resolubilised in 5 µl of 2% acetonitrile and 0.3% TFA and injected into the mass spectrometer.

Samples were loaded onto 50-cm columns packed in-house with C18 1.9 µm ReproSil particles (Dr Maisch), with an EASY-nLC 1000 system (Thermo Fisher Scientific) coupled to the MS (Q Exactive HF-X, Thermo Fisher Scientific). A homemade column oven maintained the column temperature at 60 °C. Peptides were introduced onto the column with buffer A (0.1% formic acid), and phosphosites for data-independent acquisition were eluted with a 70-min gradient starting at 3% buffer B (80% acetonitrile, 0.1% formic acid) and followed by a stepwise increase to 19% in 40 min, 41% in 20 min, 90% in 5 min and 95% in 5 min, at a flow rate of 350 nl min$^{-1}$. A data-independent acquisition MS method was used in which one full scan (300 to 1,650 m/z, R = 60,000 at 200 m/z, maximum injection time 60 ms) at a target of 3 × 10$^6$ ions was first performed, followed by 32 windows with a resolution of 30,000 where precursor ions were fragmented with higher-energy collisional dissociation (stepped collision energy 25%, 27.5%, 30%) and analysed with an AGC target of 3 × 10$^6$ ions and a maximum injection time of 54 ms in profile mode using positive polarity.

## Phosphoproteomics data processing

Mass spectrometry raw files were processed by the Spectronaut software version 14[30] using the directDIA option. Serine/threonine/tyrosine phosphorylation was added as a variable modification to the default settings, including cysteine carbamidomethylation as fixed modification and N-terminal acetylation and methionine oxidations as variable modifications. The human (reviewed 21,039, unreviewed 70,579 entries, 2015) Uniprot FASTA databases were used. The FDR was set to less than 1% at the peptide and protein levels, and a minimum length of 7 amino acids for peptides was specified. Enzyme specificity was set as C-terminal to arginine and lysine as expected using trypsin and lysC as proteases and a maximum of two missed cleavages. Data analysis was performed using Perseus software v.1.6.2.2[31]. The localization cut-off for phosphorylations was set to 0 but afterwards filtered for the localization probability of 0.75 using the Perseus plugin[32].

## NanoBit protein–protein interaction assay

N-terminal fusion constructs of NOD2 with SmBiT (11 amino acids) and LgBiT (17.6 kDa) were generated by standard cloning methods. Upon close proximity, SmBiT–LgBiT produces a strong NanoLuc luciferase signal. For protein–protein interaction assay, 2.0 × 10$^4$ wild-type and *NAGK*-knockout HEK 293T cells were seeded per well of a white opaque 96-well plate. The following day, cells were transfected in triplicates with 30 ng of each construct or control vector, respectively, using the calcium chloride and *N,N*-bis(2-hydroxyethyl)-2-amino-ethanesulfonic acid buffered saline transfection method. After 5 h incubation at 37 °C, the medium was changed, and the cells stimulated with 200 ng ml$^{-1}$ L18-MDP for 24 h. On the next day, luminescence was determined using the NanoGlo Luciferase Assay System (Promega) according to the manufacturer's instructions. The relative response ratio (RRR) was calculated by dividing the difference of the mean luminescence value of interest and the negative control by the difference of the positive

(SmBiT–PRKACA and LgBiT–PRKAR2A) and negative control (empty vector) and multiplied by 100% to receive the ratio as a percentage:

$$RRR\ (\%) = ((\text{Value of interest} - \text{negative control})/$$
$$(\text{positive control} - \text{negative control})) \times 100$$

## MTP uptake assay

Synthesis of MTP–fluorescein was performed as described previously for muramylpeptide biotinylation[33]. MurNAc-Ala-D-isoGln-Lys purchased from Invivogen and NHS Fluorescein purchased from Thermo Scientific were used as starting reagents. As a side product, lactoyl-Ala-D-isoGln-Lys-fluorescein (lac-TP-fluorescein) was also isolated. Full experimental details and structural identification data are given in the Supplementary Methods.

KBM-7 cells ($2.5 \times 10^6$ per ml) (100 µl per well final volume) were incubated with either MurNAc-Ala-D-isoGln-Lys-Fluorescein (MTP–fluorescein) or lac-TP-fluorescein (20 µM) for the indicated times at 37 °C. Cells were then washed five times with ice-cold PBS. Intracellular fluorescein levels were assessed by flow cytometry analysis on a BD LSR Fortessa.

## MDP phosphorylation for cell culture experiments and NMR analysis

A reaction mixture containing 50 mM HEPES pH 8.0, 10 mM MgCl$_2$, 50 mM NaCl, 5 mM ATP, 1.6 or 2 mM MDP with or without the addition of 3.6 µM recombinant NAGK was incubated for 2 h at 37 °C. Following incubation, reaction mixtures were heated to 95 °C for 5 min to inactivate and precipitate the recombinant NAGK. The solution was clarified by centrifugation at 20,000$g$ for 5 min, and the supernatant was collected for stimulations (reaction mixture is designated as pMDP).

## NMR analysis

The mixtures were freeze-dried and deuterium exchange was accomplished by freeze-drying sample solutions in deuterated water (D$_2$O; 99.98%, Deutero) twice. NMR spectroscopic measurements were performed at 300 K on a Bruker Avance[III] 700 MHz spectrometer equipped with an inverse 5 mm quadruple-resonance Z-grad cryoprobe (spectrometer frequencies: 700.43 MHz for $^1$H, 176.12 MHz for $^{13}$C, and 283.54 MHz for $^{31}$P). Acetone was used as an external standard for calibration of $^1$H ($\delta_H = 2.225$ ppm) and $^{13}$C ($\delta_C = 30.89$ ppm) NMR spectra; 85% of phosphoric acid was used as an external standard for calibration of $^{31}$P NMR spectra ($\delta_P = 0.00$ ppm). All data were acquired and processed by using Bruker TOPSPIN V 3.1 or higher (Bruker BioSpin Corporation). The parameter sets used were adapted starting from respective Bruker standard parameter sets, which are all included in this software. The 2D $^1$H,$^{31}$P-HMQC-TOCSY spectrum was recorded with a mixing time of 120 ms.

## ELISA

Cytokines were measured using mouse TNF and mouse IL-6 ELISA kits (BD Bioscience) or human IP-10 or human IL-8 ELISA kits (BD Bioscience) according to the manufacturer's instructions.

## Expression and purification of human NAGK

The full-length human *NAGK* gene was codon-optimized for expression in *E. coli*, purchased from IDT, and cloned into a modified pET21 vector with an N-terminal His–MBP tag. *E. coli* BL21 Rosetta (DE3) cells were cultured in 3 l TB medium until they reached $A_{600}$ of 1.0–2.0, and protein expression was induced at 37 °C with 0.3 mM IPTG for 4 h. Cell pellets were resuspended in lysis buffer (20 mM HEPES pH 7.5, 300 mM NaCl, 10 mM imidazole pH 8.0, 5% glycerol, 0.05% Triton X-100, 1 mM β-mercaptoethanol) and disrupted by sonication. Recombinant NAGK was purified over Ni-NTA affinity chromatography and washed with lysis buffer, followed by buffer HS (20 mM HEPES pH 7.5, 1 M NaCl, 10 mM imidazole pH 8.0, 5% glycerol, 1 mM β-mercaptoethanol), then buffer

A (20 mM HEPES pH 7.5, 150 mM NaCl, 10 mM imidazole pH 8.0, 5% glycerol, 1 mM β-mercaptoethanol). HIS–MBP–NAGK was eluted in buffer A supplemented with 250 mM imidazole. The His–MBP tags were subsequently removed by TEV protease cleavage at 4 °C with overnight dialysis in dialysis buffer (20 mM HEPES pH 7.5, 50 mM NaCl, 2 mM DTT). Proteins were further purified over a HiTrap Q column (GE Healthcare) and eluted with a linear gradient of buffer B (20 mM HEPES pH 7.5, 2 M NaCl, 2 mM DTT). For the final purification, fractions containing NAGK protein were loaded on a Superdex 16/60 S200 size exclusion chromatography column (GE Healthcare) in 20 mM HEPES pH 7.5, 150 mM NaCl, 5% glycerol, 1 mM TCEP. Protein samples were concentrated and flash-frozen in liquid nitrogen and stored at −80 °C.

## qPCR

RNA was isolated from $2 \times 10^6$ KBM-7 cells or $1 \times 10^6$ BMDM cells using the Total RNA Purification Mini Spin Column Kit (Genaxxon Bioscience GmbH) according to the manufacturer's protocol. The remaining DNA in the extracted RNA was digested using DNaseI (Thermo Scientific). cDNA synthesis was performed with RevertAid Reverse Transcriptase (Thermo Scientific) with Oligo(dT)18 primers. The synthesized cDNA was cleaned up using solid-phase reversible immobilization magnetic beads (GE Healthcare). For the qPCR reaction, the PowerUp SYBR Green Master Mix (Thermo Scientific) was used with the respective primer pairs. qPCR primers used in this study can be found in Supplementary Table 2.

## Western blotting

Following stimulation, cells were lysed in DISC lysis buffer (150 mM NaCl, 50 mM Tris pH 7.5, 10% glycerol, 1% Triton X-100) supplemented with cOmplete protease inhibitor cocktail (Roche) and 10 mM *N*-ethylmaleimide and then 2× SDS lysis buffer (126 mM Tris-HCl pH 8, 20% v/v glycerol, 4% w/v SDS, 0.02% w/v bromophenol blue, 5% v/v β-mercaptoethanol) and subjected to repeated freeze–boil cycles. Samples were separated using SDS–PAGE and transferred to nitrocellulose membranes. The following antibodies were used for probing: rabbit anti-NAGK (1:1,000, ab203900, Abcam), rabbit anti-RIPK2 (1:1,000, 4142, Cell Signaling Technology), mouse anti-β-actin horseradish peroxidase (HRP) (1:1,000, sc-47778, Santa Cruz Biotechnology), rabbit anti-phospho p65 (1:1,000, 3033, Cell Signaling Technology), rabbit anti-phospho p38 (1:1,000, 9211, Cell Signaling Technology), mouse anti-phospho IκBα (1:1,000, 9246, Cell Signaling Technology), mouse anti-ubiquitin (1:1,000, 3936, Cell Signaling Technology), anti-mouse IgG HRP-linked (1:5,000, 7076, Cell Signaling Technology), anti-rabbit IgG–HRP (1:5,000, 7074, Cell Signaling Technology). Source gel data are provided in Supplementary Fig. 1.

## Bioinformatics

Gene expression data was downloaded from the BioGPS homepage: http://biogps.org/downloads/ (Human U133A/GNF1H Gene Atlas–averaged values matrix). Analysis and plotting were performed in *R* (version 4.1.0.). Low variance genes ($\sigma^2$ <1) were excluded from the analysis. For the remaining genes, a Pearson correlation matrix was calculated using base R function cor(). Genes were ordered by their correlation coefficient to *NAGK*, and the cut-off was made at $r \geq 0.7$. These remaining 113 genes were used to conduct a gene ontology enrichment analysis using PANTHER GO-slim[34]. The results were ordered by significance, and a cut-off was made at FDR $\leq 10^{-7}$ (FDR was calculated by the Benjamini–Hochberg procedure). *R* code for analysis and plotting is available at Github: https://github.com/Pestudkaru/Corr_analysis.

## NAGK–NOD2 phylogenetic comparison

An iterative BLAST (PSI-BLAST) search was performed to identify homologues of NAGK using default settings (BLOSUM62 matrix; Gap costs - Existence: 11 Extension: 1; conditional compositional score matrix adjustment; PSI-BLAST threshold: 0.005). *Homo sapiens* (NCBI

AAH05371.1) and *E. coli* (NCBI WP_097585995.1) protein sequences were used as start references. All significant sequences (E value less than 0.05) were selected for subsequent PSI-BLAST rounds (up to five rounds). Sequences were uploaded to Geneious Prime to be further selected based on length (287–456 amino acids). A Clustal Omega alignment was performed to determine the conservation of human NAGK structure-based active site residues. Non-canonical isoforms and low-quality protein sequences were removed, leaving a final list of 748 NAGK protein sequences. Geneious Tree Builder with default settings was used to calculate the phylogenetic tree displayed as a clad-ogram. For NOD2 protein homologue identification, iterative BLAST (PSI-BLAST) searches were performed separately in mammals, reptiles, birds, amphibians, fish and invertebrate species. NOD2 sequences were derived and uploaded to Geneious Prime for alignment using Clustal Omega. The CARD and NACHT domains were used to discern 217 NOD2 sequences, which were then overlaid onto the NAGK tree.

## Statistics

Unless otherwise indicated, statistical significance was determined by either one-way or two-way ANOVA and a Dunnett or Šídák correction for multiple testing. The exact number of replicates (*n*) is indicated within the figure legends. Data plotting and statistical analysis were performed using GraphPad Prism 9. If several comparisons are shown with a comparison bar, the large tick of the comparison bar indicates the reference data to which the statements on the significance level refer.

## Reporting summary

Further information on research design is available in the Nature Research Reporting Summary linked to this article.

## Data availability

The mass spectrometry-based proteomics data have been deposited to the ProteomeXchange Consortium via the PRIDE partner repository and are available via the identifier PXD022384. Deep-sequencing raw data (genome-wide genetic screen) have been deposited in the NCBI Sequence Read Archive under accession number PRJNA841795. The cor-responding processed data are provided in Supplementary Table 1. Raw LC–MS data have been submitted to MassIVE and can be accessed with ID MSV000088170. Materials and reagents are available from the corre-sponding author upon request. Source data are provided with this paper.

## Code availability

R code for the analysis and plotting is available at Github: https://github.com/Pestudkaru/Corr_analysis.

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

**Acknowledgements** We thank L. Hansbauer, M. Kösters, H. Käßner and S. Thomsen for outstanding technical support; S. Suppmann and the Protein Production Core Facility (MPI, Munich) for expressing recombinant GST–UbA; BioSysM flow cytometry facility for great support; Z. Sun for helpful discussions; S. Bauernfried for providing the mScarlet targeting construct; and M. Berouti for generating figures of chemical structures. This work was supported by grants from the ERC (ERC-2020-ADG–101018672 ENGINES) and the Deutsche Forschungsgemeinschaft (DFG, German Research Foundation) CRC 1403/A03 (Project-ID 414786233) to V.H., by grants from the European Union's Framework Programme for Research and Innovation Horizon 2020 (2014–2020) under the Marie Skłodowska-Curie Grant Agreement No. 754388 and from LMU Munich's Institutional Strategy LMUexcellent within the framework of the German Excellence Initiative (no. ZUK22) to C.A.S., by grants of the Max Planck Society for the Advancement of Science, by grants from the Helmholtz Association 'ExNet-0041-Phase2-3 (SyNergy-HMGU)' and the Else Kröner Fresenius Stiftung (ForTra-gGmbH) to W.W., by grants from the DFG (Project-ID 405101514) to M.Y. and by grants from the ERC (ERC-2018-STG–804182 SOLID) to L.T.J.

**Author contributions** Conceptualization: C.A.S. and V.H. Investigation: C.A.S., A.-M.G., C.C.d.O.M., M.C.T., K.S., C.V., E.F., D.N., G.K., S.J.S., A.W., M.K.P., B.W., N.G. and T.F. Writing: C.A.S. and V.H. with input from all authors. Resources: W.W., M.Y., T.F., L.T.J., M.M. and V.H. Funding acquisition: V.H. Supervision: V.H.

**Competing interests** The authors declare no competing interests.

**Additional information**
**Correspondence and requests for materials** should be addressed to Veit Hornung.

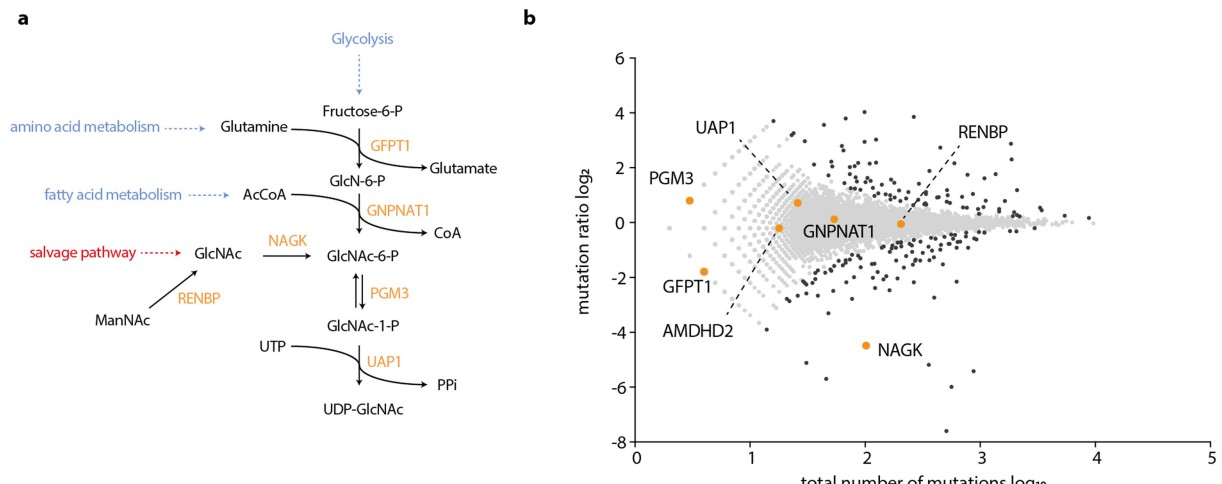

**Extended Data Fig. 1 | UDP-GlcNAc biosynthesis and salvage pathway. a**, Schematic representation of the UDP-GlcNAc biosynthesis and salvage pathway (adapted from[6]). **b**, Data from Fig. 1d with genes indicated in **a** (highlighted in orange).

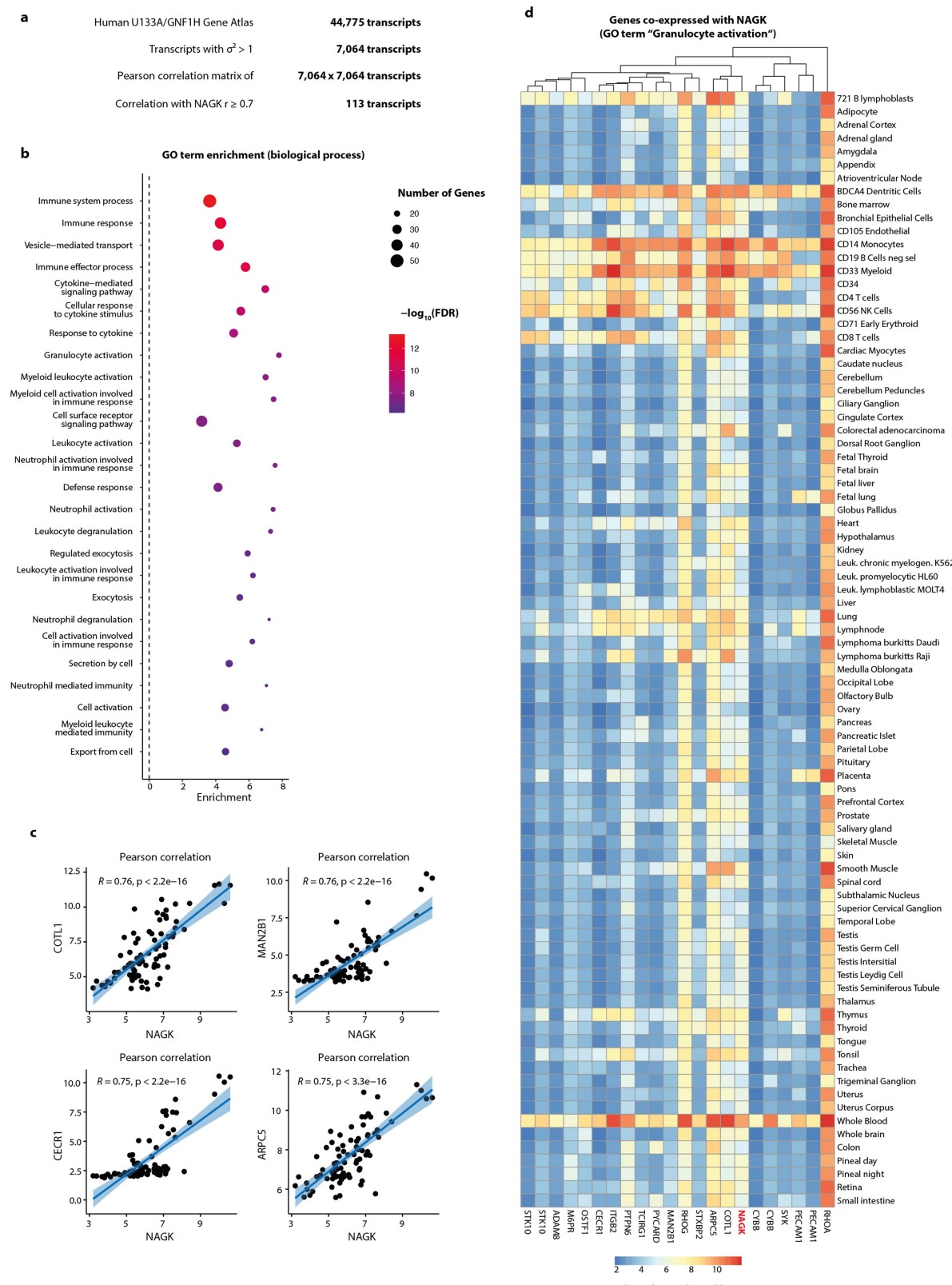

**a**

| | |
|---|---|
| Human U133A/GNF1H Gene Atlas | **44,775 transcripts** |
| Transcripts with $\sigma^2 > 1$ | **7,064 transcripts** |
| Pearson correlation matrix of | **7,064 x 7,064 transcripts** |
| Correlation with NAGK r ≥ 0.7 | **113 transcripts** |

**b** GO term enrichment (biological process)

**d** Genes co-expressed with NAGK
(GO term "Granulocyte activation")

**c** Pearson correlation

**Extended Data Fig. 2** | See next page for caption.

**Extended Data Fig. 2 | NAGK is expressed in immune cells. a**, The microarray dataset Human U133A/GNF1H Gene Atlas containing 44,775 probes was used to build a gene (Pearson) correlation matrix of 7,064 most variable transcripts ($\sigma^2 > 1$). All transcripts were ordered by their correlation to *NAGK*, and the cut-off was made at r ≥ 0.7. The remaining 113 transcripts were used to conduct the Gene Ontology analysis. **b**, Gene Ontology analysis. Most highly correlating genes to *NAGK* were used to identify over-represented biological processes. Results were plotted in order of significance as measured by the false discovery rate (FDR). Colour code from red (high significance) to violet (lower significance). The circle size indicates how many genes were found to be part of any given GO term. Fold enrichment is shown on the x-axis. **c**, Pairwise comparisons of *NAGK* and the four highest correlating genes from GO term "Granulocyte activation" with the regression line (blue) and the 95% confidence interval (light blue). Data are depicted as $\log_2$ expression values. A two tailed t-test was run on the null hypothesis that the two variables are not linearly related. No correction for multiple testing was conducted. *R* = Pearson correlation coefficient, p = P-value. **d**, Heatmap showing the expression pattern of *NAGK* together with genes from most enriched GO term "Granulocyte activation". Genes in columns are clustered by Pearson correlation. Colour from blue to red shows $\log_2$ of mean intensity values.

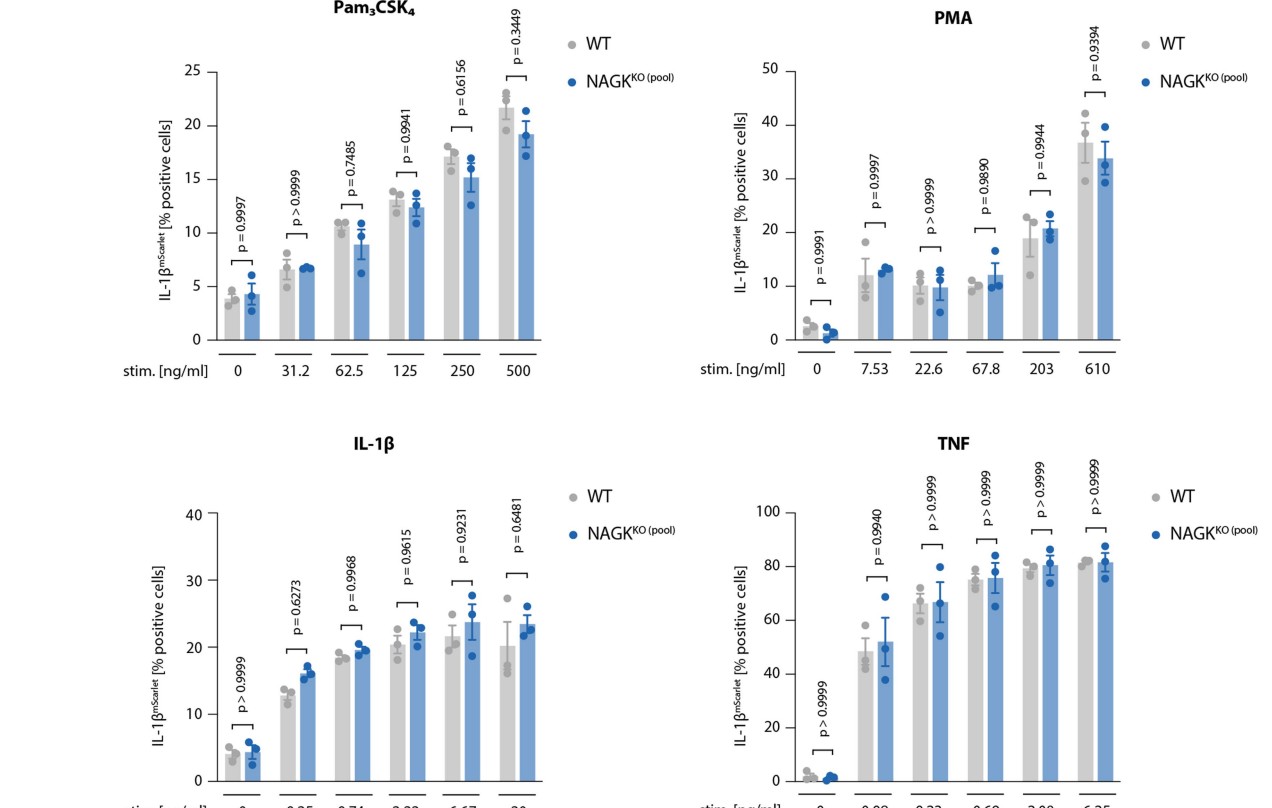

**Extended Data Fig. 3 | NAGK does not impact on other NF-kB activating immune pathways.** Flow cytometry analysis of KBM-7-IL1B$^{mScarlet}$ WT (grey) and NAGK$^{KO}$ pool (dark blue) cells treated with the indicated increasing doses of Pam$_3$CSK$_4$, PMA, recombinant IL-1β, and TNF for 16 h. Mean ± s.e.m of $n$ = 3 independent biological samples; two-way ANOVA with Šídák's multiple comparisons test. **** p < 0.0001 or as indicated.

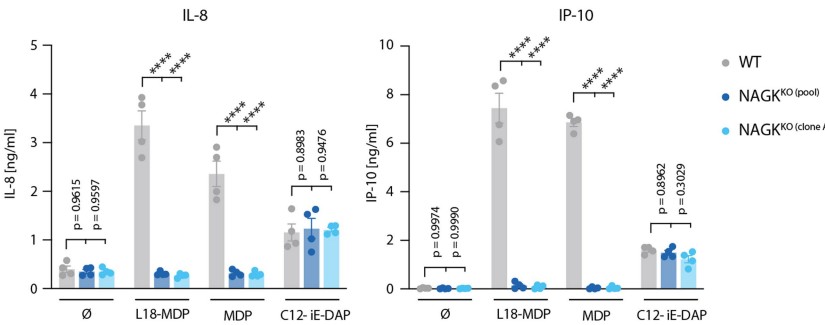

**a** Chemokine production of KBM-7 cells

**b** mRNA expression of KBM-7 cells

**Extended Data Fig. 4 | NAGK is required for the NOD2 transcriptional immune response. a**, IL-8 and IP-10 production of KBM-7-IL1B^mScarlet WT (grey), NAGK^KO pool (dark blue) and clonal (light blue) cells that were stimulated as indicated. Mean ± s.e.m of *n* = 4 independent biological samples; two-way ANOVA with Dunnet's multiple comparisons test. **b**, qPCR analysis of the indicated transcripts in KBM-7-IL1B^mScarlet WT (grey) and NAGK^KO pool (dark blue) cells treated with either MDP or iE-DAP for 4 h. Gene expression levels were normalised to the *GAPDH*. Mean ± s.e.m of *n* = 4 (n = 3 for *IFIT1*, *CXCL10*, and *NAGK*) independent biological samples; two-way ANOVA conducted on log-transformed data with Šídák's multiple comparisons test. **** p < 0.0001 or as indicated.

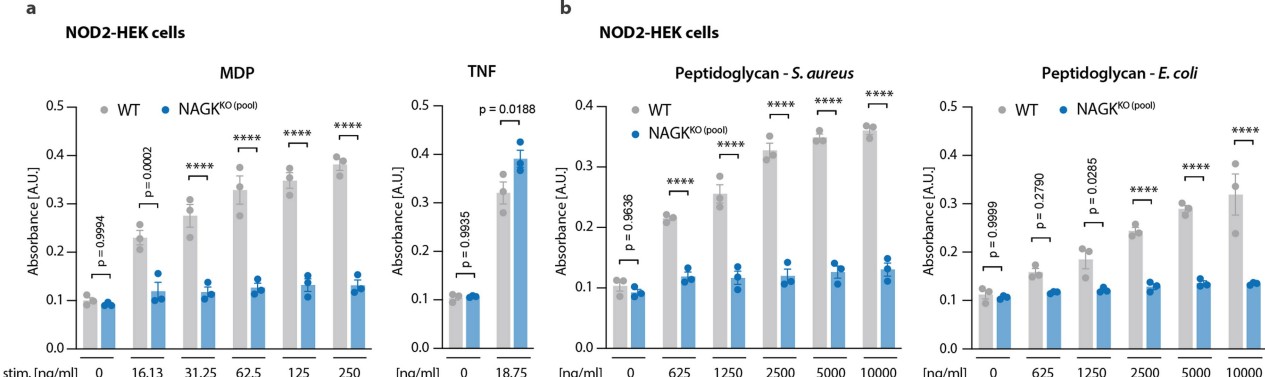

**a**

**NOD2-HEK cells**

**MDP**

**TNF**

**b**

**NOD2-HEK cells**

**Peptidoglycan - *S. aureus***

**Peptidoglycan - *E. coli***

**c**

**THP-1 cells**

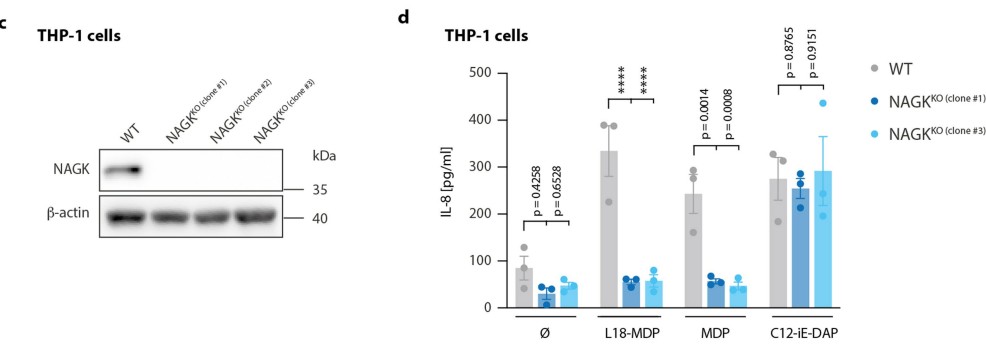

**d**

**THP-1 cells**

**e**

**KBM-7 cells**

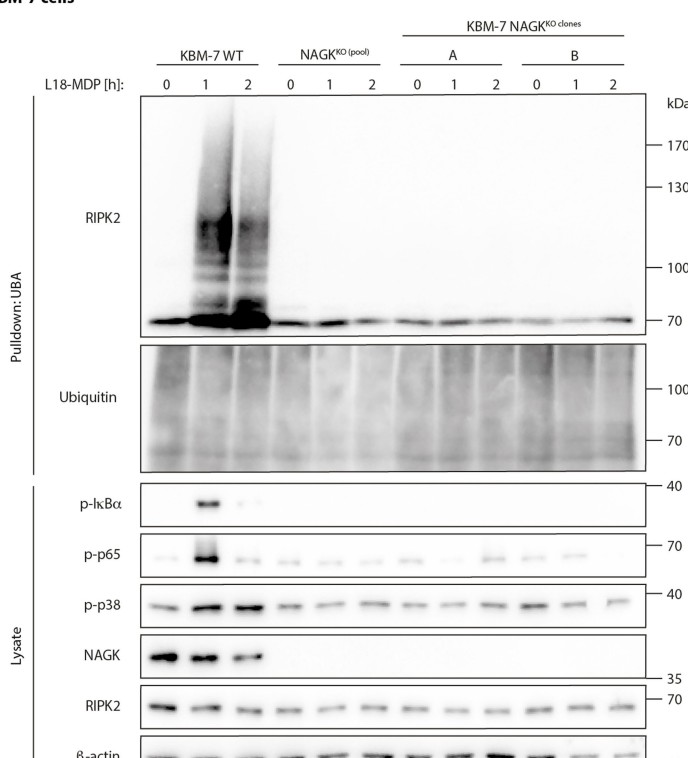

**Extended Data Fig. 5 | MDP detection in NOD2-expressing HEK cells, THP-1 cells and KBM-7 cells requires NAGK. a** and **b**, WT (grey) and NAGK^KO pool (dark blue) NOD2-HEK cells were stimulated as indicated and NF-kB activity was determined after 16 h. Mean ± s.e.m of *n* = 3 independent biological samples; two-way ANOVA with Šidák's multiple comparisons test. **c**, Immunoblotting was used to monitor NAGK levels in THP-1 WT and NAGK^KO clonal cells. (representative of three independent experiments). **d**, IL-8 production of THP-1 WT (grey), NAGK^KO clone #1 (dark blue), or clone #3 (light blue) cells was

monitored following the treatment with the indicated stimuli. Mean ± s.e.m of *n* = 3 independent biological samples; two-way ANOVA with Dunnett's multiple comparisons test. **** p < 0.0001 or as indicated. **e**, Immunoblotting was used to monitor RIPK2 ubiquitination and NF-kB and MAPK activation markers following L18-MDP stimulation in KBM-7-IL1B^mScarlet WT and NAGK^KO pool or clonal cells. Ubiquitinated proteins were enriched via pulldown of endogenous ubiquitinated protein bound to an immobilised ubiquitin-associated domain (UBA). Blot represents one of three biological replicates.

**a**

MurNAc-Ala-D-isoGln-Lys-Fluorescein (MTP-Fluorescein)

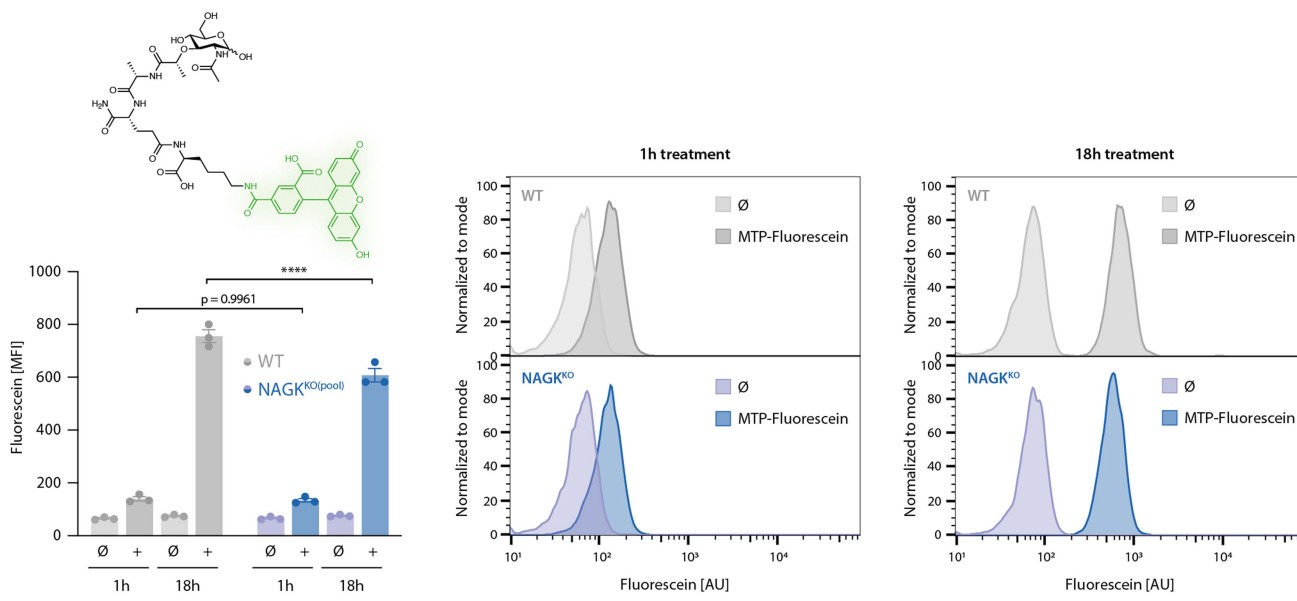

**b**

Lactoyl-Ala-D-isoGln-Lys-Fluorescein (lac-TP-Fluorescein)

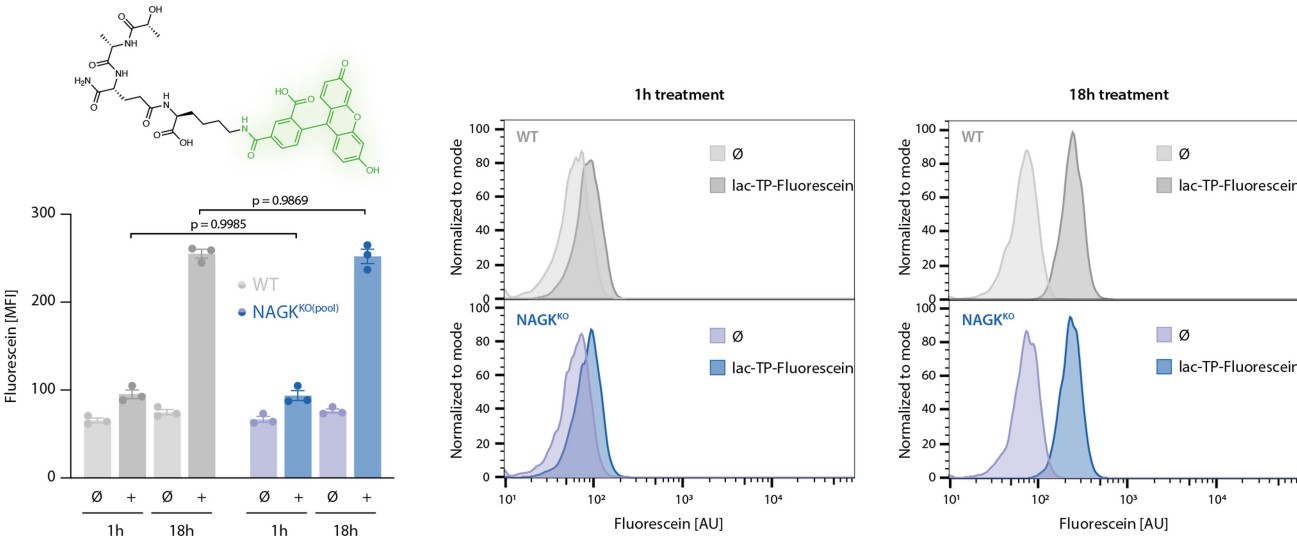

**Extended Data Fig. 6 | NAGK does not impact on MTP uptake. a** and **b**, Top left schematic representation of the Fluorescein labelled molecules. Bar charts and histograms represent flow cytometry analysis of KBM-7-IL1B$^{mScarlet}$ WT (grey) and NAGK$^{KO}$ pool (dark blue) cells treated with either MTP-Fluorescein (**a**) or lac-TP-Fluorescein (**b**) for indicated time periods, MFI represents median fluorescence intensity. Bar chart represents the mean ± s.e.m of $n$ = 3 independent biological samples; two-way ANOVA with Šídák's multiple comparisons test. Histograms represent one experiment shown of three independent experiments. The same untreated samples were used for the MTP-Fluorescein and lac-TP-Fluorescein uptake experiments as controls. **** p < 0.0001 or as indicated.

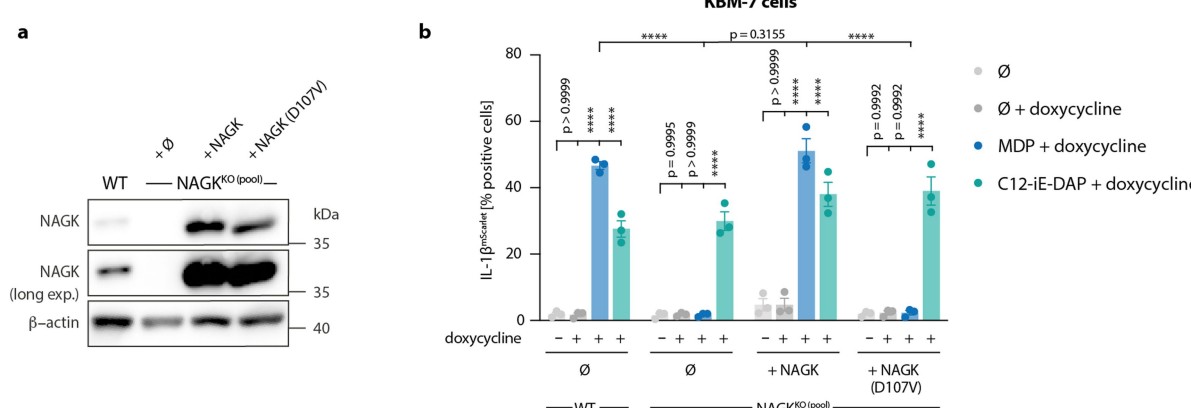

**Extended Data Fig. 7 | The kinase activity of NAGK is required upstream of NOD2. a**, Immunoblot indicating NAGK levels in KBM-7-IL1B^mScarlet WT or NAGK^KO pool cells reconstituted with WT or D107V NAGK. **b**, Flow cytometry analysis of these cells either untreated (light grey) or treated with doxycycline (dark grey) followed by MDP (dark blue) or C12-iE-DAP (green) stimulation for 16 h. Mean ± s.e.m of $n$ = 3 independent biological samples; two-way ANOVA with Dunnett's multiple comparisons test. **** $p < 0.0001$ or as indicated.

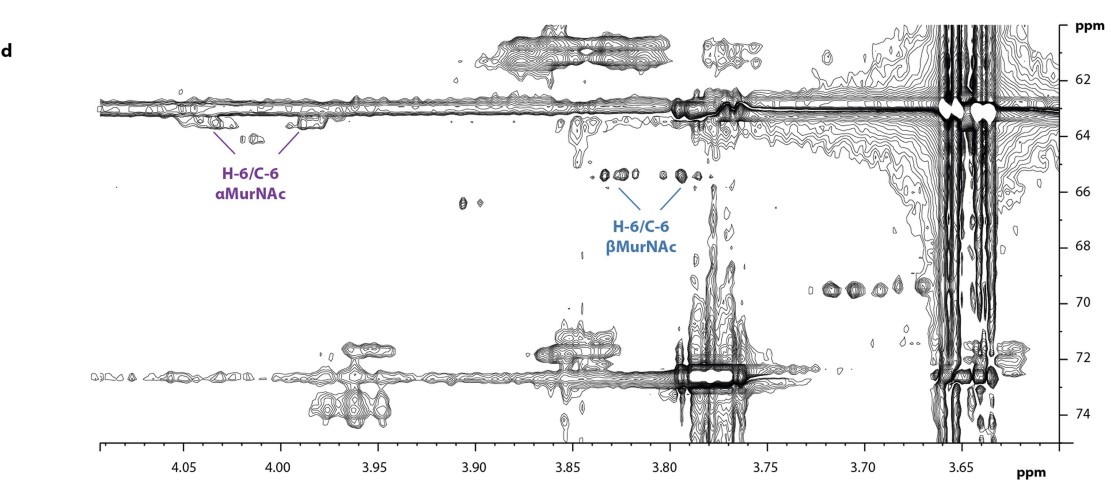

**Extended Data Fig. 8** | See next page for caption.

**Extended Data Fig. 8 | NAGK phosphorylates MDP at position O-6.** 2 mM MDP and 5 mM ATP were incubated for 2 h at 37 °C in 50 mM HEPES pH 8.0, 10 mM MgCl$_2$, and 50 mM NaCl, with or without the addition of 3.6 μM recombinant NAGK (rec. NAGK). **a**, 1D $^{31}$P NMR analysis showed the appearance of new signals ($\delta_P$ = 4.72, 4.62 ppm) deriving from monophosphates in presence of rec. NAGK, with a simultaneous, partial conversion of ATP to ADP (Fig. 3a). **b**, By 2D $^1$H,$^{31}$P-HMQC-TOCSY, the signal at $\delta_P$ 4.72 ppm could be assigned as the phosphate of the β-anomer, the signal at $\delta_P$ 4.62 ppm as the one of the α-anomer of phospho-MDP, respectively. The depicted region displays the cross correlations of the respective $^{31}$P signal to the corresponding anomeric proton (H-1) of the MurNAc residue (α: $\delta_H$ 5.18, d, $^3J_{H1,H2}$ = 3.6 Hz; β: $\delta_H$ 4.68, d, $^3J_{H1,H2}$ = 8.5 Hz). The attachment position of phosphates was determined by 2D $^1$H,$^{31}$P-HMQC (**c**) and 2D $^1$H,$^{13}$C-HSQC$_{dept}$ (**d**). For the α-anomer, the phosphate shows cross correlations to protons at $\delta_H$ 4.06-4.03 and 4.00-3.97 ppm, respectively, which are bound to a carbon at $\delta_C$ 63.5 ppm. For the β-anomer, the phosphate shows cross correlations to protons at $\delta_H$ 3.84-3.81 and 3.81-3.78 ppm, respectively, which are bound to a carbon at $\delta_C$ 65.4 ppm. This clearly indicates the phosphorylation at position O-6 of MDP.

**a**

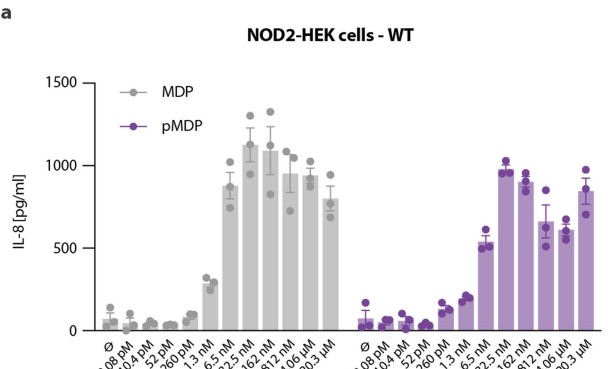

**b**

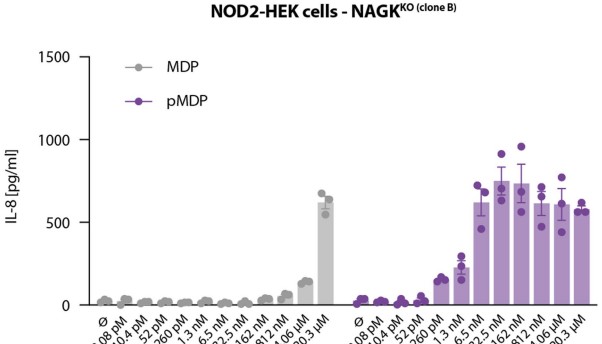

**Extended Data Fig. 9 | 6-phospho-MDP stimulates NAGK-deficient cells.** **a** and **b**, NOD2-HEK cells either WT or NAGK-deficient were treated with either MDP or pMDP in digitonin buffer at the indicated concentrations and IL-8 production was measured after 16 h by ELISA. Mean ± s.e.m of $n$ = 3 independent biological samples.

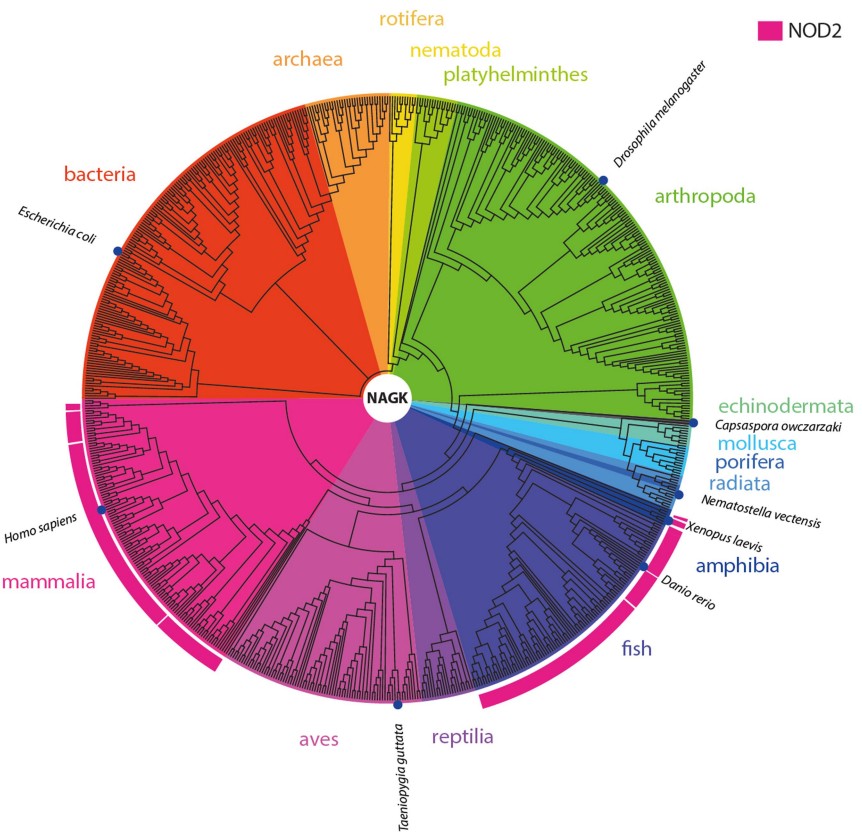

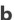

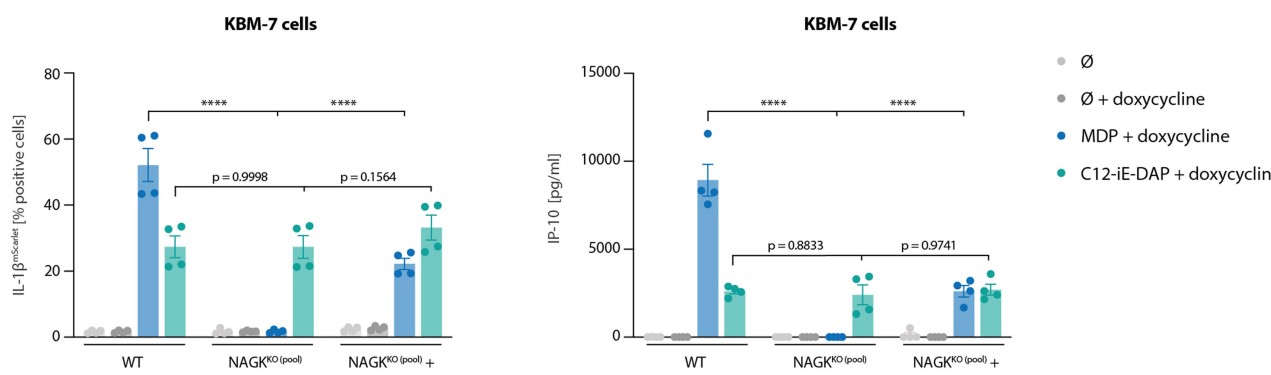

**Extended Data Fig. 10 | NAGK is conserved from prokaryotes to eukaryotes and Drosophila NAGK can operate upstream of NOD2. a**, Cladogram of NAGK proteins across the indicated species in a clockwise fashion (bacteria, archaea, rotifera, nematoda, platyhelminthes, arthropoda, echinodermata, mollusca, porifera, radiata, amphibia, fish, reptilia, aves, and mammalia). Species harbouring NOD2 protein sequences are marked in dark pink. Blue dots represent locations of specific species of interest. **b**, KBM-7-IL1B$^{mScarlet}$

WT, NAGK$^{KO}$ pool cells or NAGK$^{KO}$ pool cells reconstituted with doxycycline-inducible drosophila NAGK (CG6218) were stimulated as indicated. After 16 h they were analysed by flow cytometry or their supernatant was assessed for IP-10 production. Mean ± s.e.m of $n$ = 4 independent biological samples; two-way ANOVA with Dunnett's multiple comparisons test. **** p < 0.0001 or as indicated.

# Reporting Summary

## Statistics

For all statistical analyses, confirm that the following items are present in the figure legend, table legend, main text, or Methods section.

| n/a | Confirmed | |
|---|---|---|
| ☐ | ☒ | The exact sample size (*n*) for each experimental group/condition, given as a discrete number and unit of measurement |
| ☐ | ☒ | A statement on whether measurements were taken from distinct samples or whether the same sample was measured repeatedly |
| ☐ | ☒ | The statistical test(s) used AND whether they are one- or two-sided<br>*Only common tests should be described solely by name; describe more complex techniques in the Methods section.* |
| ☒ | ☐ | A description of all covariates tested |
| ☐ | ☒ | A description of any assumptions or corrections, such as tests of normality and adjustment for multiple comparisons |
| ☐ | ☒ | A full description of the statistical parameters including central tendency (e.g. means) or other basic estimates (e.g. regression coefficient) AND variation (e.g. standard deviation) or associated estimates of uncertainty (e.g. confidence intervals) |
| ☐ | ☒ | For null hypothesis testing, the test statistic (e.g. *F*, *t*, *r*) with confidence intervals, effect sizes, degrees of freedom and *P* value noted<br>*Give P values as exact values whenever suitable.* |
| ☒ | ☐ | For Bayesian analysis, information on the choice of priors and Markov chain Monte Carlo settings |
| ☒ | ☐ | For hierarchical and complex designs, identification of the appropriate level for tests and full reporting of outcomes |
| ☐ | ☒ | Estimates of effect sizes (e.g. Cohen's *d*, Pearson's *r*), indicating how they were calculated |

*Our web collection on statistics for biologists contains articles on many of the points above.*

## Software and code

Policy information about availability of computer code

| Data collection | For data collection the following software was used: BD FACSDiva 8.0.1 and 8.0.2. |
|---|---|
| Data analysis | All statistical analysis were performed using GraphPad Prism 9.<br><br>Flow cytometry data were analysed using FlowJo 10.7.<br><br>For correlation analysis, analysis and plotting were performed in R (version 4.1.0.). Pearson correlation matrix was calculated using base R function cor(). Genes were ordered by their correlation coefficient to NAGK, and the cut-off was made at r ≥ 0.7 (R code for analysis and plotting is available upon request).<br><br>Gene ontology enrichment analysis was done using PANTHER GO-slim.<br><br>Mass spec raw files were processed by the Spectronaut software version 14 using the directDIA option. |

For manuscripts utilizing custom algorithms or software that are central to the research but not yet described in published literature, software must be made available to editors and reviewers. We strongly encourage code deposition in a community repository (e.g. GitHub). See the Nature Portfolio guidelines for submitting code & software for further information.

## Data

Policy information about availability of data

All manuscripts must include a data availability statement. This statement should provide the following information, where applicable:

- Accession codes, unique identifiers, or web links for publicly available datasets
- A description of any restrictions on data availability
- For clinical datasets or third party data, please ensure that the statement adheres to our policy

All data including Source Data for Figs. 1–4 and Extended Data Fig. 3-7 and 9 and 10 are provided with the paper and its Supplementary Information files.

The MS-based proteomics data have been deposited to the ProteomeXchange Consortium via the PRIDE partner repository and are available via the identifier PXD022384. https://www.ebi.ac.uk/pride/archive/projects/PXD022384

Deep-sequencing raw data (genome-wide genetic screen) have been deposited in the NCBI Sequence Read Archive under accession number PRJNA841795. https://www.ncbi.nlm.nih.gov/bioproject/PRJNA841795
The corresponding processed data are provided in Supplementary Table 1.

Raw LC-MS data have been submitted to MassIVE and can be accessed with ID MSV000088170. http://dx.doi.org/10.25345/C58861

R code for analysis and plotting is available at Github: https://github.com/Pestudkaru/Corr_analysis.

Materials and reagents are available from the corresponding author upon request.

# Field-specific reporting

Please select the one below that is the best fit for your research. If you are not sure, read the appropriate sections before making your selection.

☒ Life sciences          ☐ Behavioural & social sciences          ☐ Ecological, evolutionary & environmental sciences

For a reference copy of the document with all sections, see nature.com/documents/nr-reporting-summary-flat.pdf

# Life sciences study design

All studies must disclose on these points even when the disclosure is negative.

| | |
|---|---|
| Sample size | No sample size calculation was performed, sample sizes were chosen based on previous experience and on what is common practice in the field to study essential components of this signaling cascade.<br><br>Compare: Hrdinka et al. https://doi.org/10.1016/j.celrep.2016.02.062 or Hrdinka et al. https://doi.org/10.15252/embj.201899372 or Stafford et al. https://doi.org/10.1016/j.celrep.2018.01.024 |
| Data exclusions | No data were excluded from the analysis. |
| Replication | All experiments were independently repeated as indicated in the respective figure legends. Furthermore, for most experiments, multiple cell lines were used to confirm reproducibility of the findings (KBM-7, THP-1, NOD2-HEK cells, HEK 293T and murine BMDMs). |
| Randomization | No randomization was necessary for this study, as there was no need to control for factors that would not be under direct experimental control. |
| Blinding | Blinding was not required for this study because no subjective analyses were performed that would have been biased by knowledge about the subjects studied. |

# Reporting for specific materials, systems and methods

We require information from authors about some types of materials, experimental systems and methods used in many studies. Here, indicate whether each material, system or method listed is relevant to your study. If you are not sure if a list item applies to your research, read the appropriate section before selecting a response.

## Materials & experimental systems

| n/a | Involved in the study |
|-----|----------------------|
| ☐ | ☒ Antibodies |
| ☐ | ☒ Eukaryotic cell lines |
| ☒ | ☐ Palaeontology and archaeology |
| ☐ | ☒ Animals and other organisms |
| ☒ | ☐ Human research participants |
| ☒ | ☐ Clinical data |
| ☒ | ☐ Dual use research of concern |

## Methods

| n/a | Involved in the study |
|-----|----------------------|
| ☒ | ☐ ChIP-seq |
| ☐ | ☒ Flow cytometry |
| ☒ | ☐ MRI-based neuroimaging |

# Antibodies

| | |
|---|---|
| Antibodies used | rabbit anti-NAGK (ab203900, Abcam)<br>rabbit anti-RIPK2 (#4142, Cell Signaling Technology)<br>mouse anti-β-actin HRP (sc-47778, Santa Cruz Biotechnology)<br>rabbit anti-phospho p65 (#3033, Cell Signaling Technology)<br>rabbit anti-phospho p38 (#9211, Cell Signaling Technology)<br>mouse anti-phospho IκBα (#9246, Cell Signaling Technology)<br>mouse anti-ubiquitin (#3936, Cell Signaling Technology)<br>anti-mouse IgG HRP linked (#7076, Cell Signaling Technology)<br>anti-rabbit IgG HRP (#7074, Cell Signaling Technology). |
| Validation | With the exception of the anti-NAGK antibody, we did not empirically validate the antibodies ourselves, but appropriate controls were performed to ensure that appropriate conclusions were drawn. All antibodies were purchased from commercial suppliers and the corresponding validation studies can be found on their websites:<br><br>rabbit anti-RIPK2 (#4142, Cell Signaling Technology)<br>https://www.cellsignal.com/products/primary-antibodies/rip2-d10b11-rabbit-mab/4142<br><br>mouse anti-β-actin HRP (sc-47778, Santa Cruz Biotechnology)<br>https://www.scbt.com/p/beta-actin-antibody-c4<br><br>rabbit anti-phospho p65 (#3033, Cell Signaling Technology)<br>https://www.cellsignal.com/products/primary-antibodies/phospho-nf-kb-p65-ser536-93h1-rabbit-mab/3033<br><br>rabbit anti-phospho p38 (#9211, Cell Signaling Technology)<br>https://www.cellsignal.com/products/primary-antibodies/phospho-p38-mapk-thr180-tyr182-antibody/9211<br><br>mouse anti-phospho IκBα (#9246, Cell Signaling Technology)<br>https://www.cellsignal.com/products/primary-antibodies/phospho-ikba-ser32-36-5a5-mouse-mab/9246<br><br>mouse anti-ubiquitin (#3936, Cell Signaling Technology)<br>https://www.cellsignal.com/products/primary-antibodies/ubiquitin-p4d1-mouse-mab/3936<br><br>anti-mouse IgG HRP linked (#7076, Cell Signaling Technology)<br>https://www.cellsignal.com/products/secondary-antibodies/anti-mouse-igg-hrp-linked-antibody/7076<br><br>anti-rabbit IgG HRP (#7074, Cell Signaling Technology)<br>https://www.cellsignal.com/products/secondary-antibodies/anti-rabbit-igg-hrp-linked-antibody/7074 |

# Eukaryotic cell lines

Policy information about cell lines

| | |
|---|---|
| Cell line source(s) | KBM-7 cells were from Thijn Brummelkamp.<br>THP-1 cells were from DSMZ (ACC 16).<br>HEK 293T cells were from DSMZ (ACC 635).<br>NOD2-HEK cells (HEK-Blue™-NOD2 cells) were from Invivogen. |
| Authentication | Cell lines were not additionally authenticated. |
| Mycoplasma contamination | All cell lines were tested negative for mycoplasma contamination at the beginning of the study, but were not tested routinely thereafter. |
| Commonly misidentified lines<br>(See ICLAC register) | No commonly misidentified cell lines were used in this study. |

# Animals and other organisms

Policy information about studies involving animals; ARRIVE guidelines recommended for reporting animal research

| | |
|---|---|
| Laboratory animals | 1 female and 2 male WT mice (C57BL/6J) and 2 female and 1 male Nagk-/- mice (C57BL/6J) (all littermates) aged 12 weeks were used for BMDM isolation. |
| | All mice were housed in standard cages in a specific pathogen-free facility (21±1°C, on a 12-h light/dark cycle, with average humidity of around 55%) with ad libitum access to food and water in the animal facility at the Centre for Neuropathology. |
| Wild animals | No wild animals were used in this study. |
| Field-collected samples | No field-collected samples were used in this study. |
| Ethics oversight | All mice were handled according to institutional guidelines approved by the animal welfare and use committee of the government of Upper Bavaria. |

Note that full information on the approval of the study protocol must also be provided in the manuscript.

# Flow Cytometry

## Plots

Confirm that:

☒ The axis labels state the marker and fluorochrome used (e.g. CD4-FITC).

☒ The axis scales are clearly visible. Include numbers along axes only for bottom left plot of group (a 'group' is an analysis of identical markers).

☒ All plots are contour plots with outliers or pseudocolor plots.

☒ A numerical value for number of cells or percentage (with statistics) is provided.

## Methodology

| | |
|---|---|
| Sample preparation | Cells were collected and washed with PBS. Cells were then passed through a 40-μm cell strainer and then vortexed before analysis. |
| Instrument | Cells were analysed on a BD LSR Fortessa and cells were sorted on a BD Fusion cell sorter (both BD Biosciences). |
| Software | Flow cytometry data were analysed using FlowJo 10.7. For data collection the following software was used: BD FACSDiva 8.0.1 and 8.0.2. |
| Cell population abundance | Conclusions regarding cell population abundance were conducted when measuring the expression of the mScarlet reporter in KBM-7 cells. For these studies, a customised gate was drawn based on negative control samples (unstimulated controls) and then applied to all other samples of the respective data set. |
| Gating strategy | For analytical flow cytometry: Cells were first gated on FSC-A vs SSC-A to exclude debris. Subsequently, single cells were gated using the FSC-H vs FSC-W plot. Single cells were then analysed for fluorescence positivity by adjusting the threshold of the gate to non-fluorescent cells. Cell sorting of fixed cells for mutation mapping: cells were identified using the FCS-A vs SSC-A blot to exclude debris. Subsequently, single cells were identified using DAPI-A vs DAPI-W blot. Single cells were displayed in a histogram, on which gating on 1n DNA content was performed to exclude diploid cells. Sorting was performed on the haploid cell population, the gates used for sorting were set to the bottom 4% IL1B-mScarlet-low and top 4% IL1B-mScarlet-high cells. |

☒ Tick this box to confirm that a figure exemplifying the gating strategy is provided in the Supplementary Information.

