## [Peer Review File · Nature]

Manuscript Title: Phosphorylation of muramyl peptides by NAGK is required for NOD2 activation

Reviewer Comments & Author Rebuttals

Reviewer Reports on the Initial Version:

Referees' comments:

Referee #1 (Remarks to the Author):

Stafford et al have identified a novel requirement for NOD2 signaling by showing that a kinase named NAGK phosphorylates MDP on the C6 of the muramyl group and that this modification is essential for NOD2-mediated detection of MDP.

Overall, this is a relatively straightforward study and the data provided are generally convincing within the somewhat reductionist context of the authors' approach. However, while the data could be potentially of general interest to the field of innate immunity and are certainly unexpected and novel, the overall significance of the findings remains uncertain. In particular:

1. Data are limited to cancer cell lines and in particular the relatively uncommon cell line KBM-7, which was originally used for the genome-wide screen in Fig. 1. Those cells are chromosomally very peculiar (near haploid) and certainly do not represent physiological settings. Most data also rely on a rather artificial readout (IL1b-mScarlet %positive cell) that is difficult to fully evaluate (why expressing everything as % positive cells?). More data showing cytokine readout (such as IL-8 ELISA in Fig. 2c) should be provided.
2. Evidence in NAGK knockout mice and primary cells from KO mice (MEFs/BMDMs...) would be essential to validate the importance of the findings and the physiological relevance.
3. All the data are using MDP as an agonist. While this is fine for initial investigation, this should be complemented by using polymeric peptidoglycan or muramidase-digested peptidoglycan, since this represents more faithfully what cells encounter. Proving that NAGK is essential in response to PGN is critical.
4. There is no proof here that MDP is phosphorylated in vivo or in cellulo by NAGK. Following incubation of cells with MDP or PGN, MDP should be converted to p-MDP in a NAGK-dependent manner. Proving that p-MDP can be retrieved following incubation of WT cells but not NAGK KO cells with MDP is the only faithful way to prove that NAGK is a true MDP kinase in vivo.
5. There are also no data showing a role for NAGK in the context of an infection with live bacteria. While it is not always easy to isolate a NOD2-dependent effect on inflammatory signaling in cells infected with whole bacteria (because of redundancy of pathways), it remains feasible for instance in epithelial cells engineered to knockout NOD1. Thus, cellular responses to bacterial infection in NOD1 KO versus NOD1/NAGK DKO would allow characterization of the role of NAGK in the context of an infection with an intracellular bacterial pathogen, which is lacking so far here. I think this would strengthen the overall claim and the manuscript.
6. The fact that L18-MDP also requires NAGK opens some questions for which the authors do not have clear answers: indeed, since both the L18 and the phospho group are on the same C6 of MDP, this suggests that L18-MDP must be first converted to MDP within the cytosol by an unknown

esterase before NOD2 could detect the p-MDP. However, this remains a speculation at this stage as there is no proof that L18-MDP is converted to MDP. Moreover, this unknown esterase should be essential for NOD2 signaling and should have been identified in the genome-wide screen, unless there is functional redundancy.

Moreover, some specific comments below should be considered as they are important for the interpretation of the presented data:

-For Fig. 3g-h, given the prime importance of these results for the general conclusions raised by the authors, I think data with the NAGK KO clone rather than a pool should be provided for KBM-7 cells. They have the clone as it appears in Fig. 2c. In addition, measurement of IL8 by ELISA as in Fig. 2c should be done, in addition to this % positive cells shown.

-Fig. 3g-h: Control experiments with NOD1 ligand should be added to control for any effect relating to endocytosis of the ligands. TNF cannot control for this. Ideally, the authors would want to use two distinct NOD1 ligands that vary on the presence or not of the MurNAc (L-Ala-D-Glu-mesoDAP versus MurNAc-L-Ala-D-Glu-mesoDAP) since, although both ligands stimulate NOD1, the MurNAc has an effect on internalization by endocytosis (see PMID 19570976).

-Fig. 3g: I don't understand why the response to p-MDP is reduced by approx.. 50% in NAGK KO cells. This is puzzling and I think a titration of the dose-response to MDP vs p-MDP in WT/NAGK KO cells would be critical.

Minor:

-“As expected, IL1BmScarlet expression following iE-DAP treatment, a specific NOD1 agonist, or TNF stimulation, were both independent of NOD2 (Fig. 1c).”. Technically, this is incorrect since there was more induction of the reporter system after TNF stimulation in NOD2 KO cells (Fig. 1c, significant difference: ***). This needs to be explained or statement revised.

-Some figures were truncated on the right side on the PDF I received.

-Figure 2b: L18-MPD should be L18-MDP

Referee #2 (Remarks to the Author):

In this manuscript Stafford et al. have used a forward genetic screen in a cell line (KBM-7) in an attempt to identify novel pieces of the NOD2 signaling pathway. They have unexpectedly identified N-acetylglucosamine kinase (NAGK) as a protein required for sensing of muramyl dipeptide (MDP), a fragment of bacterial peptidoglycan previously identified as the minimal fragment that activates NOD2. The screening data are convincing, and the authors follow up the finding by directly making cell lines (KBM-7, THP-1, HEK) deficient in NAGK which replicate a defect in recognizing MDP. The authors show that NAGK can phosphorylate n-acetylglucosamine in synthetic MDP and that the kinase activity of NAGK is necessary for NOD2-mediated sensing of MDP. Overall, these data are convincing. However, I have several concerns with the manuscript as presented.

Major concerns:

First, the important conclusion of the study is that the “real” ligand for NOD2 is NAGK-phosphorylated MDP. However, direct binding information is not provided. If this manuscript is going to adjust the definition of the NOD2 ligand, it will be important to see directly, for example, that MDP is a poor ligand for NOD2 (low affinity), while NAGK-phosphorylated MDP binds better (high affinity). Some structural discussion of why this modification binds better to NOD2 would also be good but may not be essential at this stage.

Second, NOD2, of course, does not naturally sense chemically synthesized MDP. It senses bacteria, and bacterial peptidoglycan is diversely modified by many enzymes. For example, peptidoglycan O-acetyltransferase acetylates, as I understand it, the very site identified here as the target of NAGK. Perhaps that (or some other similarly naturally occurring version of peptidoglycan) is the “natural” ligand for NOD2, while NAGK is only important for sensing synthetic MDP. The authors made a minor attempt at addressing purified natural peptidoglycan in Figure 2 regarding NOD2-overexpressing HEK cells, but this is insufficient. It is essential to determine whether NAGK is necessary for sensing of bacteria by naturally expressed NOD2.

Minor comments:

Are there any indications that NAGK is important for immune sensing of infection? For example, are there polymorphisms in NAGK that, like NOD2, relate to disease risk?

If, as suggested, an important role of phosphorylation of n-acetylglucosamine in the cytosol by NAGK is to make it charged and less membrane permeable, do the authors understand why phosphorylating MDP with NAGK before adding it to cell culture didn't disrupt its ability to cross a membrane and interact with a cytosolic sensor?

As is, the data are based entirely on cell lines. Data with primary cells would strengthen the study.

UW forwarding additional comments to the editor: .. this adjustment to the definition of the NOD2 ligand does not change much in the bigger picture. NOD2 still senses peptidoglycan or peptidoglycan products in the cytosol. What is different about how we think about mechanisms of innate immunity after knowing that NAGK phosphorylates MDP? There's no new immunological paradigm and no disease is now better understood.

Referee #3 (Remarks to the Author):

In this paper Stafford et al study the sensing of muramyl dipeptide (MDP) by Nucleotide-binding oligomerization domain 2 (NOD2) using a genetic screen. NOD2 is a cytosolic pattern recognition receptor implicated in the sensing of peptidoglycan, a component of the bacterial cell wall. Activation of NOD2 leads to stimulation of the NFκB pathway and genetic defects in NOD2 have been linked to Crohn's disease.

Stafford et al modify haploid human KBM7 cells to contain a T2A-mScarlet fluorescent marker in the NFκB target gene IL1B and demonstrate that this results in a NOD2-dependent induction in fluorescence in response to MDP or lipid-modified MDP (L18-MDP). Subsequently they use insertional mutagenesis in order to identify genes required for NOD2-dependent MDP sensing in a haploid cell line. As expected, components of the NFκB pathway were detected, whereas the SLC15A family transporter molecules previously implicated in sensing endosomally-derived MDP (Nakamura et al, Nature, 2014) were not. In addition, they identify NAGK, a kinase implicated in hexosamine biosynthesis. Focusing on NAGK, which has not been previously implicated in pathogen sensing, they demonstrate that NAGK does not affect the machinery downstream of NOD2 but rather influences its dimerization upon MDP exposure. Inspired by the resemblance of MDP to GlcNAc, the authors demonstrated that NAGK could phosphorylate MDP and, importantly, effective sensing of MDP in NAGK-KO cells could be bypassed by in vitro phosphorylation of MDP prior to exposure.

Together, this study delivers a single, clear message: MDP sensing by NOD2 is facilitated by NAGK. Although it is not conceptually new that host factors modify pathogenic factors that get sensed, the high medical burden of bacterial infections could make this message important and in addition this work has the potential to address outstanding mechanistic issues related to NOD2 ligand binding.

Comments:

1. Although unexpected, this manuscript presents a relatively straightforward finding that does not change the current view that NOD2 senses peptidoglycan. One could wonder if this warrants publication in Nature. Furthermore, in comparison to publications in Nature, the data-density is relatively low: an unusual amount of space is used for introduction and conclusions, half of the experimental work describes what NAGK does not do and the conclusion could be summarized in a single figure: screen (A), validation and specificity for NOD2 ligand (B), phosphorylation of MDP by NAGK (C) and modified MDP signals in NAGK-KO cells (D).
2. I do not understand the statement “Structural studies suggest that MDP can directly bind and activate NOD2 (11), yet direct evidence in support of this model is lacking.” Reference 11 shows that NOD2 has a potential ligand binding site but does not show that MDP can directly bind to NOD2. Some studies that were not cited actually study how MDP binds NOD2: Schaefer et al, ACS Chem Biol 2017, Leimkuhler Grimes et al, J. Am. Chem. Soc. 2012. Thus, although there may be uncertainty in the field, this is not addressed by Stafford et al. Ideally, one would like to see how MDP binds to NOD2 (in a structure) and the authors should examine the affinity of MDP and phosphorylated MDP for NOD2.
3. Although this work shows that NAGK facilitates sensing of MDP by NOD2, it has not been demonstrated if this is relevant in the context of pathogen exposure. This may seem likely but for various reasons the relevance could be limited: would unmodified MDP also bind (and activate) NOD2, and if so, what is the PAMP concentration encountered during infection? Although NAGK facilitates sensing it would be important to know the affinities of modified and non-modified MDP for NOD2. Is MDP the only bacterial product that gets sensed by NOD2? This work is only relevant if

the importance of MDP sensing by NOD2 has been verified. Would NAGK affect the concentration, stability or presentation of intracellular MDP? In this scenario NAGK may not be needed to respond to a larger (polyvalent) peptidoglycan structure that may be more stable and may be less membrane-permeable. Thus, more data is required to establish that the finding is important in the context of pathogen exposure.

Additional questions:

-Could the authors address why the SLC15A peptide transporters were not identified (redundancy or the use of L18-MDP?).

-The authors show that NAGK can phosphorylate MDP in vitro but not that this occurs in cells.

-What are all the other significant hits in the screen and will the authors provide all sequencing data and analysis scripts?

Author Rebuttals to Initial Comments:

Above all, we would like to thank the reviewers for their valuable time in evaluating our manuscript in great detail. We now present a thoroughly revised manuscript that takes into account all critical points raised by the reviewers. Moreover, we have even added additional new experiments that further improve the quality of the manuscript (see list below). We believe that the manuscript has been significantly improved by considering these comments and suggestions, and we would like to sincerely thank the reviewers for taking the time to review our manuscript and providing us with valuable comments and suggestions.

Overview of changes and additions:

Figure	Additional data
Fig. 1	b: changed x-axis from bi-exponential to exponential
Fig. 2	Rearranged and added new figures: Removed c and d plots and replaced with IL-8 ELISA data for NOD2-HEK cells treated with MDP and different peptidoglycans.
Fig. 3	Rearranged and added new figures: c: Combined data from previous Extended Data Fig. 7 to generate new figure d: Performed new in vitro kinase data and corresponding mass spectrometry e, f: Performed in cellulo mass spec (metabolomics) to determine NAGK dependent phosphorylation of MDP and conversion of L18-MDP to MDP h: Performed new bypass experiment with optimised delivery protocol in KBM-7s h, i: Performed new bypass experiment with precise dose titrations in NOD2-HEK cells
Fig. 4	New figure: Isolated BMDMs from newly generated Nagk ^{-/-} mice treated with MDP or Pam ₃ CSK ₄ showing that NAGK is also essential for NOD2 signalling in primary cells
Extended Data Fig. 1	Unchanged
Extended Data Fig. 2	Unchanged
Extended Data Fig. 3	Unchanged
Extended Data Fig. 4	a: Moved KBM-7 IL-8 and IP-10 ELISA data from previous Fig. 2c to this figure.
Extended Data Fig. 5	New figure: NF-kB activity assays of dose titrations of MDP and purified bacterial peptidoglycans on NOD2-HEK cells
Extended Data Fig. 6	THP-1 cytokine data from previous Extended Data Fig. 5 was moved here.

Extended Data Fig. 7	KBM-7 signalling data moved from previous Extended Data Fig. 6 was moved here.
Extended Data Fig. 8	New figure: Performed uptake FACS analysis of Fluorescein modified MTP (NOD2 ligand) on WT and NAGK KO KBM-7s, displaying NAGK not having an impact on uptake.
Extended Data Fig. 9	Moved kinase mutant data from previous Fig. 3b to this figure, showing kinase activity of NAGK essential for NOD2 signalling.
Extended Data Fig. 10	New figure: NMR analysis of phosphorylated MDP, indicating phosphorylation on carbon 6 of the sugar moiety
Extended Data Fig. 11	New figure: Dose titration of MDP and pMDP in NOD2-HEK cells using IL-8 ELISA as a readout, indicating that pMDP can bypass NAGK deficiency.
Extended Data Fig. 12	NAGK evolution cladogram from previous Extended Data Fig. 8 was moved here.
Extended Data Fig. 13	New figure: a, Multiple alignment of human, mouse, drosophila and bovine NAGK indicating conserved regions. b, Overexpression of drosophila NAGK (CG6218) indicating that it can recover the NAGK KO phenotype in KBM-7 cells
Supplementary Table 1	New table that provides the raw data of the screening data presented in Fig. 1.

Response to the reviewers' comments

Referee #1 (Remarks to the Author):

Stafford et al have identified a novel requirement for NOD2 signaling by showing that a kinase named NAGK phosphorylates MDP on the C6 of the muramyl group and that this modification is essential for NOD2-mediated detection of MDP.

Overall, this is a relatively straightforward study and the data provided are generally convincing within the somewhat reductionist context of the authors' approach. However, while the data could be potentially of general interest to the field of innate immunity and are certainly unexpected and novel, the overall significance of the findings remains uncertain. In particular:

1. Data are limited to cancer cell lines and in particular the relatively uncommon cell line KBM-7, which was originally used for the genome-wide screen in Fig. 1. Those cells are chromosomally very peculiar (near haploid) and certainly do not represent physiological settings.

We agree with the reviewer that KBM7 cells are not necessarily the most physiological cell model. As such, we have also validated the role of NAGK for NOD2 signalling in additional cell lines. We show that THP-1 cells (Fig. 2f, g and Extended Data Fig. 6), as well as NOD2-expressing HEK cells (Fig. 2c-e and Extended Data Fig. 5) depend on NAGK for their NOD2 response. Most importantly, we now have also generated *Nagk*^{-/-} mice (Fig. 4), of which we isolated bone marrow-derived macrophages. Here, we measured a broad array of different readouts, which are summarised in detail in point 2 below.

Most data also rely on a rather artificial readout (IL1b-mScarlet %positive cell) that is difficult to fully evaluate (why expressing everything as % positive cells?). More data showing cytokine readout (such as IL-8 ELISA in Fig. 2c) should be provided

In our hands, the readout of mScarlet % positive cells nicely correlates with pro-inflammatory gene expression. As such, given its ease of use and low cost, we used it as our standard readout for the engineered KBM7 cell line. We have now changed Fig. 1b to better indicate the gating strategy. Moreover, we have also added more data, in which we study cytokine/chemokine production as a readout of NOD2 stimulation (e.g. Fig. 2d, e and Fig. 4a and Extended Data Fig. 11).

2. Evidence in NAGK knockout mice and primary cells from KO mice (MEFs/BMDMs...) would be essential to validate the importance of the findings and the physiological relevance.

In agreement with this comment, we generated *Nagk*^{-/-} mice through targeted CRISPR/Cas9 deletion of exons 4 – 5 of the *Nagk* gene (Fig. 4). Consistent with our data obtained from immortalised cell lines, the generated bone marrow derived macrophages from these mice were unable to respond to NOD2 activation. We used the following readouts to study their response: ELISA to measure TNF or IL-6, qPCR to measure a broad range of inflammatory and antiviral markers, ubiquitin pulldowns to measure RIPK2 ubiquitination, immunoblots to study NF-κB and MAPK activation as well as a global view of activation using phosphoproteomics. These data clearly indicate the essential requirement of NAGK in NOD2 signalling in more physiologically relevant cells. Of note, NOD2-independent stimuli trigger normal pro-inflammatory gene expression in the absence of NAGK.

3. All the data are using MDP as an agonist. While this is fine for initial investigation, this should be complemented by using polymeric peptidoglycan or muramidase-digested peptidoglycan, since this represents more faithfully what cells encounter. Proving that NAGK is essential in response to PGN is critical.

The reviewer is correct that our previous manuscript had mainly focused on MDP, and not on other PGN breakdown products or on natural PGN species. We also found that muramyl-tripeptide (Lys-MTP) requires NAGK for its NOD2-stimulatory capacity (data not shown). Since it is impossible to account for all possible PGN-breakdown products, we additionally conducted experiments with whole peptidoglycan (PGN) preparations from both *S. aureus* (insoluble) and *E. coli* (soluble) that we tested for their activity in NOD2-expressing HEK cells (e.g. Fig. 2e and Extended Data Fig. 5). Of note, this cell model was used because it allows us to study NOD2 in a reductionist setting without the activity of the NOD1 and/or TLR2 pathways. At increasing concentrations of the PGN in WT cells, we observed a robust increase in both IL-8 production, as well as the NF-κB-reporter assay. However, the NAGK-deficient cells were unable to respond to either PGN. Taken together, these stimuli represent a more complex physiological ligand for NOD2 stimulation that should encompass all possible breakdown products that arise within the cell. Since these complex PGN preparations also require NAGK to exert NOD2 stimulatory activity, one can conclude that NAGK is generally required for NOD2 stimulation (and not only for MDP).

4. There is no proof here that MDP is phosphorylated in vivo or in cellulo by NAGK. Following incubation of cells with MDP or PGN, MDP should be converted to p-MDP in a NAGK-dependent manner. Proving that p-MDP can be retrieved following incubation of WT cells but

not NAGK KO cells with MDP is the only faithful way to prove that NAGK is a true MDP kinase *in vivo*.

We completely agree that the *in cellulo* detection for the conversion of MDP to p-MDP in a NAGK-dependent manner would provide unequivocal evidence that NAGK is indeed an essential kinase for NOD2 activation. We therefore employed sensitive mass spectrometry (metabolomics pipeline) on L18-MDP treated WT and NAGK-deficient KBM-7 cells (Fig. 3e, f). We successfully detected L18-MDP within treated cells and also observed the conversion of L18-MDP to MDP in both WT and NAGK-deficient cells. The presence of p-MDP was completely dependent on NAGK, supporting the essential role of NAGK *in cellulo*. Interestingly, we also observed higher amounts of MDP in NAGK-deficient cells, which argues for an accumulation of this metabolite in the absence of NAGK.

5. There are also no data showing a role for NAGK in the context of an infection with live bacteria. While it is not always easy to isolate a NOD2-dependent effect on inflammatory signaling in cells infected with whole bacteria (because of redundancy of pathways), it remains feasible for instance in epithelial cells engineered to knockout NOD1. Thus, cellular responses to bacterial infection in NOD1 KO versus NOD1/NAGK DKO would allow characterization of the role of NAGK in the context of an infection with an intracellular bacterial pathogen, which is lacking so far here. I think this would strengthen the overall claim and the manuscript.

In line with the reviewer's comments, we strived to provide a more physiologically relevant NOD2 stimulation compared to synthetic MDP to ensure the general applicability of our finding. As outlined in point 3, we used two different complex peptidoglycan preparations and studied their activity in cells, in which only NOD2-dependent sensing is operational. These results showed that PGN recognition was completely NAGK-dependent.

The reviewer is right that it would be great to establish infection conditions, in which one could isolate the contribution of NOD2 and/or NAGK specifically. However - as also noted by the reviewer - this would be a rather difficult venture *in vivo*, in that there is a redundancy with NOD1-dependent and also TLR-dependent sensing pathways (and maybe other PRRs). Our comprehensive set of data, which is now based on many different cell lines and stimulations, clearly indicates that NAGK deficiency fully phenocopies a NOD2 deficiency. As such, we believe that the informative value of crossing *Nagk*^{-/-} mice to other PRR-deficient mice would be limited at this point.

6. The fact that L18-MDP also requires NAGK opens some questions for which the authors do not have clear answers: indeed, since both the L18 and the phospho group are on the same C6 of MDP, this suggests that L18-MDP must be first converted to MDP within the cytosol by an unknown esterase before NOD2 could detect the p-MDP. However, this remains a speculation at this stage as there is no proof that L18-MDP is converted to MDP. Moreover, this unknown esterase should be essential for NOD2 signaling and should have been identified in the genome-wide screen, unless there is functional redundancy.

The reviewer raises an important point that we had not conclusively addressed in our first manuscript. We now provide additional data, in which we studied the abundance of L18-MDP, MDP and also p-MDP using mass spectrometry *in cellulo*. These results clearly indicate that L18-MDP is converted to MDP *in cellulo*, which liberates the OH group of the C6 of MDP for phosphorylation (Fig. 3e, f). We assume that the loss of this acyl group is mediated by carboxylesterases, as it has been shown for other prodrugs of this kind (for a review please see

PMID: 30245959). Of note, given the fact that these esterases act redundantly to each other, we do not expect to find them as hits in a forward genetic screen of this kind.

Of note, to dispel any doubt that MDP is phosphorylated at the OH group of the C6 position, we additionally conducted NMR studies, in which we identified this exact modification (Extended Data Fig. 10).

Moreover, some specific comments below should be considered as they are important for the interpretation of the presented data:

-For Fig. 3g-h, given the prime importance of these results for the general conclusions raised by the authors, I think data with the NAGK KO clone rather than a pool should be provided for KBM-7 cells. They have the clone as it appears in Fig. 2c. In addition, measurement of IL8 by ELISA as in Fig. 2c should be done, in addition to this % positive cells shown.

We restructured this panel to also address another important point that was raised by this reviewer during the review process (see below question “*Fig. 3g: I don’t understand why the response to p-MDP is reduced by approx.. 50% in NAGK KO cells....*”). To this end, we repeated the “p-MDP bypass” experiment with digitonin delivery instead of adding the compounds to the cell culture medium. This approach led to a dramatically increased potency of MDP and p-MDP, indicating that membrane passage constitutes a major bottleneck for these compounds. We now also conducted these experiments in a NAGK-deficient cell clone (Fig. 3h, i) instead of the pool KO cell line, as requested by the reviewer. Conducting a dose titration experiment to determine EC₅₀ values, we found that p-MDP displays a slightly increased potency than MDP in WT cells (EC₅₀ values of 0.119 nM vs. 0.461 nM respectively). When we delivered these compounds into NAGK KO cells, the potency of p-MDP remained comparable (EC₅₀ value of 0.134 nM), whereas MDP dramatically dropped in its activity. To this end, the potency of MDP dropped by 12,619-fold compared to p-MDP. While this experiment did show NAGK-independent NOD2 activity, we would consider these compound concentrations unrealistically high, considering the fact that the cells had to be transiently permeabilized with digitonin.

For these set of experiments, we also determined IL-8 production by ELISA, as requested (Extended Data Fig. 11).

-Fig. 3g-h: Control experiments with NOD1 ligand should be added to control for any effect relating to endocytosis of the ligands. TNF cannot control for this. Ideally, the authors would want to use two distinct NOD1 ligands that vary on the presence or not of the MurNAc (L-Ala-D-Glu-mesoDAP versus MurNAc-L-Ala-D-Glu-mesoDAP) since, although both ligands stimulate NOD1, the MurNAc has an effect on internalization by endocytosis (see PMID 19570976).

This is an interesting point which we did not consider. In the revised version we have now direct and indirect evidence that would argue against a role of NAGK for MDP uptake.

Foremost and most importantly, following the advice given by the reviewer, we delivered MDP and p-MDP into cells using the digitonin approach established in Lee at al. (PMID 19570976). This method bypasses endocytic uptake mechanisms by directly delivering these compounds into the cytoplasm. These experiments confirmed the critical role of NAGK in NOD2 stimulation (Fig. 3h, i).

Further, our new mass spectrometry data show that L18-MDP and MDP are found in cells treated with L18-MDP independently of NAGK (Fig. 3e, f). In fact, NAGK-deficient cells display slightly increased MDP levels, which is likely due to the fact that MDP cannot be converted to p-MDP. Again, these data argue against a role for NAGK in regulating PGN or MDP uptake.

Finally, we also conducted uptake studies to control for this possibility. To this end, we used Lys-MTP (MurNAc-Ala-D-isoGln-Lys), in which the distal ϵ -amino function can serve as a target group for fluorescein labelling (Extended Data Fig. 8a, b). Fluorescein-labeled Lys-MTP signals in a NAGK-dependent manner (P2P Fig. 1) and its uptake is not compromised in NAGK-deficient cells. Interestingly, at a late time point (18 hours) we see slightly more Fluorescein signal in WT cells than in NAGK-deficient cells. We assume that this is due to the fact that NAGK-dependent phosphorylation of Lys-MTP results in a negatively charged molecule that is less likely to pass the plasma membrane.

P2P Fig. 1: WT or NAGK-deficient KBM-7-IL1B^{mScarlet} cells were left untreated or incubated with MTP-Fluorescein (1 μ g/ml) for 18 hours. Subsequently, mCherry expression was measured as a proxy for NF- κ B activation. Mean \pm s.e.m of n = 3.

-Fig. 3g: I don't understand why the response to p-MDP is reduced by approx.. 50% in NAGK KO cells. This is puzzling and I think a titration of the dose-response to MDP vs p-MDP in WT/NAGK KO cells would be critical.

The reviewer raises an important point that also puzzled us, but which we have now meticulously dissected in the course of the review process: In the previous set of experiments, we had used an in vitro kinase preparation, in which MDP was not fully converted to p-MDP. Consequently, we had used a mixture of MDP / p-MDP to conduct the bypass experiment. We have come to realize that p-MDP is far less membrane permeable due to its negative charge, hence its potency is lower than MDP when directly added to the medium. The reason why WT cells responded better to the previous in vitro kinase derived p-MDP/MDP combination is because they were stimulated with both MDP and p-MDP, the former intracellularly being converted to p-MDP by NAGK. In NAGK-deficient cells, however, MDP cannot be converted to p-MDP and all NOD2 agonistic activity was ascribable to the p-MDP added to the cells. Since p-MDP is less membrane permeable, this resulted in lower NF- κ B activity in NAGK-deficient cells. Finally, the reason why we had not seen this phenomenon in HEK-NOD2 cells is that we had stimulated these cells with saturating amounts of MDP/p-MDP so that uptake was not relevant anymore.

With our new in vitro kinase preparation, we now see full conversion of MDP to p-MDP (see also our new Fig. 3d to illustrate this). To circumvent the problem of differential uptake, we conducted these experiments using digitonin-based membrane permeabilization as suggested by the reviewer. Moreover, we conduct a broad titration curve to determine EC₅₀ values. These new experiments show that p-MDP is slightly more potent than MDP in WT cells and that p-

MDP retains full activity in NAGK-KO cells, whereas the potency of MDP drops by more than 4 logs (12.619-fold).

Minor:

-“As expected, IL1BmScarlet expression following iE-DAP treatment, a specific NOD1 agonist, or TNF stimulation, were both independent of NOD2 (Fig. 1c).”. Technically, this is incorrect since there was more induction of the reporter system after TNF stimulation in NOD2 KO cells (Fig. 1c, significant difference: ***). This needs to be explained or statement revised.

Thank you for pointing this out. We now revised this statement accordingly.

-Some figures were truncated on the right side on the PDF I received.

We apologize for this mishap. The formatting has now been corrected.

-Figure 2b: L18-MPD should be L18-MDP

This has been corrected.

Referee #2 (Remarks to the Author):

In this manuscript Stafford et al. have used a forward genetic screen in a cell line (KBM-7) in an attempt to identify novel pieces of the NOD2 signaling pathway. They have unexpectedly identified N-acetylglucosamine kinase (NAGK) as a protein required for sensing of muramyl dipeptide (MDP), a fragment of bacterial peptidoglycan previously identified as the minimal fragment that activates NOD2. The screening data are convincing, and the authors follow up the finding by directly making cell lines (KBM-7, THP-1, HEK) deficient in NAGK which replicate a defect in recognizing MDP. The authors show that NAGK can phosphorylate n-acetylglucosamine in synthetic MDP and that the kinase activity of NAGK is necessary for NOD2-mediated sensing of MDP. Overall, these data are convincing. However, I have several concerns with the manuscript as presented.

Major concerns:

First, the important conclusion of the study is that the “real” ligand for NOD2 is NAGK-phosphorylated MDP. However, direct binding information is not provided. If this manuscript is going to adjust the definition of the NOD2 ligand, it will be important to see directly, for example, that MDP is a poor ligand for NOD2 (low affinity), while NAGK-phosphorylated MDP binds better (high affinity). Some structural discussion of why this modification binds better to NOD2 would also be good but may not be essential at this stage.

We agree with the reviewer that more biochemical insight into p-MDP recognition would indeed be interesting. However, in light of previously published work, we note that binding assays have thus far not shown a correlation between binding and agonistic activity. For example, work by Grimes and co-workers has shown that various MDP derivatives bind to a

recombinant NOD2 construct with K_d values in the low nanomolar range (PMID: 28708377). However, these experiments also revealed that there was no correlation between NOD2-binding and NOD2-agonistic activity as determined by NF- κ B reporter assays. As such, several compounds that displayed high affinity NOD2-binding were completely inactive in stimulating NOD2-expressing cells. In another study (PMID: 27748583), in which the solitary NOD2 LRR was probed for MDP binding, similar results were observed: The NOD2-LRR bound to 6-amino-MDP, but also to 6-amino-GlucNAc as well as the dipeptide only. In a third study by the same group (PMID: 22857257) the biologically active isomer MDP-D and inactive isomer MDP-L were immobilised onto a self-assembled monolayer for SPR measurements of full-length NOD2. When conducting SPR measurements at different pH conditions, they observed that NOD2 had the propensity to display increased interaction with MDP (both active and inactive forms of MDP) as the pH was lowered. As mentioned above, it is inexplicable that they found the K_d for the biologically active isomer of MDP to be 51, whilst the biologically inactive isomer still bound to NOD2 with a K_d of 150 nM. They also utilised full length NOD2 (pI >6) for this assay, so that the net charge of NOD2 would be positive at the pH used (pH 5.5). As they indicate in their supplementary material, NOD2 interacts with the chip independently of MDP at pH 6.5. Unfortunately, no raw pH 5.5 control chip SPR data was provided within the manuscript, nor in the supplemental data, so we can only speculate that the positive NOD2 molecule at pH 5.5 would lead to NOD2 interaction with the chip after the 10-minute time point (utilised for all readings at this pH). They do state that they subtract non-specific binding of NOD2 from all measurements, therefore their data could indicate that NOD2 can bind MDP independently of a phosphorylation on the C6 of the sugar moiety. Our latest data would be in-line with this finding, where we utilise digitonin to provide exceedingly high doses of MDP to the cell (Fig 3 h, i). Indeed, our data does indicate that NOD2 can be activated without the phosphorylation of MDP, albeit with drastically lower activity compared to the phosphorylated MDP. Therefore, the approach Grimes utilises where large amounts of MDP are coupled to the surface of the chip could result in a localised high concentration of MDP. This localised high concentration could facilitate NOD2 interaction with MDP, and would not be relevant in a physiological context where ligand availability within the cytoplasm is limited.

Despite the potential shortcomings of the papers highlighted above, we agree with the reviewer that providing biochemical evidence for the binding of pMDP to recombinant NOD2 would be beneficial for the manuscript and the field. To this end, we teamed up with the group of Prof. Geyer (Univ. of Bonn) who helped us to conduct SPR-based binding experiments. We generated two biotinylated NOD2 constructs for these studies: We obtained rabbit NOD2 lacking the N-terminal CARDS (E188-L1013) from either Sf9 insect cells (Sf9 ocNOD2) or HEK293T cells (HEK 293T ocNOD2) and immobilised them onto streptavidin functionalised sensor chips. We utilised two pH buffers for our assays (pH 5.5 and 7.4) to attempt to recapitulate the pH dependent binding observed by Grimes et al 2012.

Doing so, we were unable to observe MDP or pMDP binding at pH 7.4 (P2P Fig. 2). As hinted to above, when using pH 5.5 buffer alone, we observed significant drift of the response (RU) (P2P Fig. 3). We suspect this is due to the pH being below the theoretical pI of the protein (full-length NOD2 used by Grimes et al. or our constructs have a pI above 6). The pH 5.5 buffer would therefore cause the protein to have a net positive charge, leading to potential interaction with the chip and observed drift independent of a ligand. Nevertheless, on top of the observed drift we indeed observed an increased response to high concentrations of pMDP (black lines) when compared to MDP (blue lines, P2P Fig. 4). However, we do not believe that these data represent the true binding potential of NOD2, as these results would indicate K_d values in the

μM range. This is in stark contrast to the EC_{50} values obtained in our cell assays, which are in the pM range (Fig. 3h, i).

[Redacted]

[Figure Redacted]

[Figure Redacted]

[Redacted]

[Figure Redacted]

[Redacted]

We speculate that either NOD2 requires additional cofactors in the form of post-translational modifications (e.g. phosphorylation), membrane interaction (such as curvature), or protein interaction cofactors, although we would expect such an interactor to be indicated in our screen.

We of course agree with the reviewer that structural studies of full length NOD2 bound to either MDP or p-MPD would be very informative. As mentioned above, we speculate that we will need NOD2 in a yet-to-be-defined “functional state” and hence we now try to work with full length NOD2 isolated from signalling-competent cells. However, we hope that the reviewer agrees with us that it is clearly beyond the scope of this report, in which the role of NAGK and its function were first described, to provide structural data on the interaction of NOD2 with p-MDP.

Second, NOD2, of course, does not naturally sense chemically synthesized MDP. It senses bacteria, and bacterial peptidoglycan is diversely modified by many enzymes. For example, peptidoglycan O-acetyltransferase acetylates, as I understand it, the very site identified here as the target of NAGK. Perhaps that (or some other similarly naturally occurring version of peptidoglycan) is the “natural” ligand for NOD2, while NAGK is only important for sensing synthetic MDP. The authors made a minor attempt at addressing purified natural peptidoglycan in Figure 2 regarding NOD2-overexpressing HEK cells, but this is insufficient. It is essential to determine whether NAGK is necessary for sensing of bacteria by naturally expressed NOD2.

The author is correct that our previous manuscript had mainly focused on MDP, and not on natural PGN species. We conducted additional experiments with muramyl-tripeptide (MTP), which also requires NAGK for its NOD2-stimulatory capacity (P2P Fig. 1, please see page 6 above). Since it is impossible to account for all possible PGN-breakdown products, we additionally conducted experiments with whole peptidoglycan (PGN) preparations from both *S. aureus* and *E. coli* (Fig. 2e and Extended Data Fig. 5). At increasing concentrations of the PGN in WT cells, we observed a robust increase in both IL-8 cytokine production as well as the NF- κ B-reporter assay. The NAGK-deficient cells were unable to respond to either type of PGN. Taken together, these stimuli represent a more complex physiological ligand for NOD2 stimulation that should encompass all possible breakdown products that arise within the cell. Importantly, these complex PGN preparations also require NAGK to exert NOD2 stimulatory activity. As such, one can conclude from these data that NAGK is generally required for NOD2 stimulation (and not only for MDP).

Furthermore, the author correctly points out that most Gram-positive bacteria encode for peptidoglycan O-acetyltransferases (e.g. OatA in *S. aureus*), which acetylate the exact same

OH group that is phosphorylated by NAGK. This modification is not complete (not all residues are affected) and it is also not all too stable (pH sensitive). Nevertheless, it is well-established that OatA serves as a virulence factor, in that O-acetylation of the muramoyl residues within peptidoglycan greatly increases its resistance to lysozyme-dependent degradation (compare PMID: 15661003). Interestingly, it has also been shown that *S. aureus* lacking OatA (containing PGN with muramoyl residues that have unmodified C6 OH groups) also displays higher pro-inflammatory activity (PMID: 28943328). These observations fit well our narrative of the importance of this OH group for NAGK phosphorylation and as such NOD2 recognition (indeed, one could argue that OatA serves to generate “stealth PGN” that cannot be phosphorylated by NAGK anymore). However, we have to be cautious in inferring this causality, since lysozyme-dependent PGN degradation is a prerequisite for NOD2 stimulation in the first place. As such, we cannot separate the role of C6 OH acetylation for lysozyme- or NAGK-resistance. We now additionally discuss this point in the manuscript.

Apart from these considerations, we would like to point out that exchanging this OH group for an amino group completely blunted the NOD2-agonistic activity of MDP. This again underscores the importance for NAGK-dependent phosphorylation of this group for NOD2-dependent recognition.

Minor comments:

Are there any indications that NAGK is important for immune sensing of infection? For example, are there polymorphisms in NAGK that, like NOD2, relate to disease risk?

To our knowledge, there is no monogenic disease associated with a defect in NAGK. However, according to the GWAS catalogue, there is an association of a SNP (rs2160783) within the NAGK gene region with changes in eosinophil counts. This particular SNP constitutes an eQTL for NAGK, as documented by the GTEx Portal. As such, one can attribute differential NAGK expression to this genetic variant. While peripheral blood eosinophilia is observed in certain subgroups of Crohn’s disease patients (PMID: 29112200), it is of course speculative whether these observations indeed constitute a causal connection.

That being said, however, we would respectfully point out that many genes that have been shown to have a clear causal relationship with certain innate immune signalling cascades may not necessarily be the cause of genetic diseases. For example, there are no known genetic diseases associated with the signalling molecules MAVS or PYCARD (ASC), although their central function in innate immunity is well established.

If, as suggested, an important role of phosphorylation of n-acetylglucosamine in the cytosol by NAGK is to make it charged and less membrane permeable, do the authors understand why phosphorylating MDP with NAGK before adding it to cell culture didn’t disrupt its ability to cross a membrane and interact with a cytosolic sensor?

We indeed, believe that this is the case. The previous bypass experiment had suggested that extracellular p-MDP is indeed just as active as MDP in WT cells, hence arguing against the idea that phosphorylation of MDP decreases membrane permeability. However, we have come to realize that the previous result was confounded by the fact that the in vitro kinase reaction contained a mixture of MDP and p-MDP. Indeed, in KBM-7 NAGK KO cells, adding the p-MDP/MDP mixture resulted in decreased activity, which we ascribe to the lower membrane permeability of p-MDP (NAGK KO cells cannot respond to MDP). WT cells, in contrast, could also respond to MDP by converting it intracellularly to p-MDP. Of note, in HEK cells, MDP

and p-MDP had shown similar activities since we had stimulated these cells in the plateau phase.

We have now conducted the bypass experiments using transient membrane permeabilization with digitonin to exclude this problem (Fig. 3g-i).

As is, the data are based entirely on cell lines. Data with primary cells would strengthen the study.

In agreement with this comment, we generated NAGK-deficient mice through targeted CRISPR/Cas9 deletion of exons 4 – 5 of the *Nagk* gene (Fig. 4). Consistent with our immortalised cell data, the generated bone marrow derived macrophages from these mice were unable to respond to NOD2 activation. We used the following readouts to observe the lack of signal: ELISA to measure TNF or IL-6, ubiquitin pulldowns to measure RIPK2 ubiquitination, qPCR to measure a broad range of inflammatory and antiviral markers, as well as a global view of activation using phosphoproteomics. These data clearly indicate the essential requirement of NAGK in NOD2 signalling in more physiologically relevant cells.

UW forwarding additional comments to the editor: .. this adjustment to the definition of the NOD2 ligand does not change much in the bigger picture. NOD2 still senses peptidoglycan or peptidoglycan products in the cytosol. What is different about how we think about mechanisms of innate immunity after knowing that NAGK phosphorylates MDP? There's no new immunological paradigm and no disease is now better understood.

We kindly disagree with this notion. In our opinion, uncovering the role of a previously poorly characterized enzyme as an essential step in NOD2 activation indeed constitutes a paradigm shift. After all, MDP is the “oldest” chemically defined PAMP molecule that is commonly used as the textbook example of PRR-dependent PAMP recognition. Please see below an excerpt of the current edition of the *Janeway Immunobiology* textbook (Fig. 3.17), in which PAMPs and their PRRs are explained. Our study now completely changes the picture of this sensing modality. A conceptually intriguing aspect of our finding is that NAGK appears to be an evolutionary remnant of the bacterial cell wall recycling pathway that is well conserved in eukaryotes. As such, NOD2 must have evolved to sense phospho-MDP in the first place and not MDP. We now include additional data, in which we show that *Drosophila* NAGK (gene name: CG6218) can complement for human NAGK, despite the fact that PGN recognition in *Drosophila* relies on an entirely different system (Extended Data Fig. 13).

In this context, we would also like to point out that NOD2 constitutes one of medically most relevant PRR pathways.

Referee #3 (Remarks to the Author):

In this paper Stafford et al study the sensing of muramyl dipeptide (MDP) by Nucleotide-binding oligomerization domain 2 (NOD2) using a genetic screen. NOD2 is a cytosolic pattern recognition receptor implicated in the sensing of peptidoglycan, a component of the bacterial cell wall. Activation of NOD2 leads to stimulation of the NF κ B pathway and genetic defects in NOD2 have been linked to Crohn's disease.

Stafford et al modify haploid human KBM7 cells to contain a T2A-mScarlet fluorescent marker in the NF κ B target gene IL1B and demonstrate that this results in a NOD2-dependent induction in fluorescence in response to MDP or lipid-modified MDP (L18-MDP). Subsequently they use insertional mutagenesis in order to identify genes required for NOD2-dependent MDP sensing in a haploid cell line. As expected, components of the NF κ B pathway were detected, whereas the SLC15A family transporter molecules previously implicated in sensing endosomally-derived MDP (Nakamura et al, Nature, 2014) were not. In addition, they identify NAGK, a kinase implicated in hexosamine biosynthesis. Focusing on NAGK, which has not been previously implicated in pathogen sensing, they demonstrate that NAGK does not affect the machinery downstream of NOD2 but rather influences its dimerization upon MDP exposure. Inspired by the resemblance of MDP to GlcNAc, the authors demonstrated that NAGK could phosphorylate MDP and, importantly, effective sensing of MDP in NAGK-KO cells could be bypassed by in vitro phosphorylation of MDP prior to exposure.

Together, this study delivers a single, clear message: MDP sensing by NOD2 is facilitated by NAGK. Although it is not conceptionally new that host factors modify pathogenic factors that get sensed, the high medical burden of bacterial infections could make this message important and in addition this work has the potential to address outstanding mechanistic issues related to NOD2 ligand binding.

Comments:

1. Although unexpected, this manuscript presents a relatively straightforward finding that does not change the current view that NOD2 senses peptidoglycan. One could wonder if this warrants publication in Nature. Furthermore, in comparison to publications in Nature, the data-density is relatively low: an unusual amount of space is used for introduction and conclusions, half of the experimental work describes what NAGK does not do and the conclusion could be summarized in a single figure: screen (A), validation and specificity for NOD2 ligand (B), phosphorylation of MDP by NAGK (C) and modified MDP signals in NAGK-KO cells (D).

We kindly disagree with the notion that a certain density threshold for data would be required for publication in *Nature*. Our aim is to provide high quality and clear data that allows the reader to easily comprehend and follow the findings. We have now performed important additional experiments, including the generation of a NAGK-deficient mouse which has led to a more comprehensive and detailed manuscript.

2. I do not understand the statement "Structural studies suggest that MDP can directly bind and activate NOD2 (11), yet direct evidence in support of this model is lacking." Reference 11 shows that NOD2 has a potential ligand binding site but does not show that MDP can directly bind to NOD2. Some studies that were not cited actually study how MDP binds NOD2: Schaefer et al, ACS Chem Biol 2017, Leimkuhler Grimes et al, J.Am. Chem. Soc. 2012. Thus, although there may be uncertainty in the field, this is not addressed by Stafford et al. Ideally, one would like to see how MDP binds to NOD2 (in a structure) and the authors should examine the affinity of MDP and phosphorylated MDP for NOD2.

We apologise for the wording of our initial submission. We were aware of these manuscripts, although we were cautious to draw strong conclusions from these data, due to the uncertainty within the field and a lack of NOD2 structural data bound to MDP. We have rewritten the manuscript and altered our discussion.

We agree with the reviewer that more biochemical insight into p-MDP recognition would indeed be interesting. However, in light of previously published work, we note that binding assays have thus far not shown a correlation between binding and agonistic activity. For example, work by Grimes and co-workers has shown that various MDP derivatives bind to a recombinant NOD2 construct with K_d values in the low nanomolar range (PMID: 28708377). However, these experiments also revealed that there was no correlation between NOD2-binding and NOD2-agonistic activity as determined by NF- κ B reporter assays. As such, several compounds that displayed high affinity NOD2-binding were completely inactive in stimulating NOD2-expressing cells. In another study (PMID: 27748583), in which the solitary NOD2 LRR was probed for MDP binding, similar results were observed: The NOD2-LRR bound to 6-amino-MDP, but also to 6-amino-GlucNAc as well as the dipeptide only. In a third study by the same group (PMID: 22857257) the biologically active isomer MDP-D and inactive isomer MDP-L were immobilised onto a self-assembled monolayer for SPR measurements of full-length NOD2. When conducting SPR measurements at different pH conditions, they observed that NOD2 had the propensity to display increased interaction with MDP (both active and inactive forms of MDP) as the pH was lowered. As mentioned above, it is inexplicable that they found the K_d for the biologically active isomer of MDP to be 51, whilst the biologically inactive isomer still bound to NOD2 with a K_d of 150 nM. They also utilised full length NOD2 (pI >6) for this assay, so that the net charge of NOD2 would be positive at the pH used (pH 5.5). As they indicate in their supplementary material, NOD2 interacts with the chip independently of MDP at pH 6.5. Unfortunately, no raw pH 5.5 control chip SPR data was provided within the manuscript, nor in the supplemental data, so we can only speculate that the positive NOD2 molecule at pH 5.5 would lead to NOD2 interaction with the chip after the 10-minute time point (utilised for all readings at this pH). They do state that they subtract non-specific binding of NOD2 from all measurements, therefore their data could indicate that NOD2 can bind MDP independently of a phosphorylation on the C6 of the sugar moiety. Our latest data would be in-line with this finding, where we utilise digitonin to provide exceedingly high doses of MDP to the cell (Fig 3 h, i). Indeed, our data does indicate that NOD2 can be activated without the phosphorylation of MDP, albeit with drastically lower activity compared to the phosphorylated MDP. Therefore, the approach Grimes utilises where large amounts of MDP are coupled to the surface of the chip could result in a localised high concentration of MDP. This localised high concentration could facilitate NOD2 interaction with MDP, and would not be relevant in a physiological context where ligand availability within the cytoplasm is limited.

Despite the potential shortcomings of the papers highlighted above, we agree with the reviewer that providing biochemical evidence for the binding of pMDP to recombinant NOD2 would be beneficial for the manuscript and the field. To this end, we teamed up with the group of Prof. Geyer (Univ. of Bonn) who helped us to conduct SPR-based binding experiments. We generated two biotinylated NOD2 constructs for these studies: We obtained rabbit NOD2 lacking the N-terminal CARDs (E188-L1013) from either Sf9 insect cells (Sf9 ocNOD2) or HEK293T cells (HEK 293T ocNOD2) and immobilised them onto streptavidin functionalised sensor chips. We utilised two pH buffers for our assays (pH 5.5 and 7.4) to attempt to recapitulate the pH dependent binding observed by Grimes et al 2012.

Doing so, we were unable to observe MDP or pMDP binding at pH 7.4 (P2P Fig. 2). As hinted to above, when using pH 5.5 buffer alone, we observed significant drift of the response (RU)

(P2P Fig. 3). We suspect this is due to the pH being below the theoretical pI of the protein (full-length NOD2 used by Grimes et al. or our constructs have a pI above 6). The pH 5.5 buffer would therefore cause the protein to have a net positive charge, leading to potential interaction with the chip and observed drift independent of a ligand. Nevertheless, on top of the observed drift we indeed observed an increased response to high concentrations of pMDP (black lines) when compared to MDP (blue lines, P2P Fig. 4). However, we do not believe that these data represent the true binding potential of NOD2, as these results would indicate K_d values in the μM range. This in stark contrast to the EC_{50} values obtained in our cell assays, which are in the pM range (Fig. 3h, i).

[Redacted]

[Figure Redacted]

[Figure Redacted]

[Redacted]

[Redacted]

[Figure Redacted]

We speculate that either NOD2 requires additional cofactors in the form of post-translational modifications (e.g. phosphorylation), membrane interaction (such as curvature), or protein interaction cofactors, although we would expect such an interactor to be indicated in our screen.

We of course agree with the reviewer that structural studies of full length NOD2 bound to either MDP or p-MDP would be very informative. As mentioned above, we speculate that we will need NOD2 in a yet-to-be-defined “functional state” and hence we now try to work with full length NOD2 isolated from signalling-competent cells. However, we hope that the reviewer agrees with us that it is clearly beyond the scope of this report, in which the role of NAGK and its function were first described, to provide structural data on the interaction of NOD2 with p-MDP.

3. Although this work shows that NAGK facilitates sensing of MDP by NOD2, it has not been demonstrated if this is relevant in the context of pathogen exposure. This may seem likely but for various reasons the relevance could be limited: would unmodified MDP also bind (and activate) NOD2, and if so, what is the PAMP concentration encountered during infection? Although NAGK facilitates sensing it would be important to know the affinities of modified and non-modified MDP for NOD2. Is MDP the only bacterial product that gets sensed by NOD2? This work is only relevant if the importance of MDP sensing by NOD2 has been verified. Would NAGK affect the concentration, stability or presentation of intracellular MDP? In this scenario NAGK may not be needed to respond to a larger (polyvalent) peptidoglycan structure that may be more stable and may be less membrane-permeable. Thus, more data is required to establish that the finding is important in the context of pathogen exposure.

The reviewer is correct that our previous manuscript had mainly focused on MDP, and not on natural PGN species. We also found that muramyl-tripeptide (Lys-MTP) requires NAGK for its NOD2-stimulatory capacity (P2P Fig. 1, please see page 6 above). Since it is impossible to account for all possible PGN-breakdown products, we additionally conducted experiments with whole peptidoglycan (PGN) preparations from both *S. aureus* (insoluble) and *E. coli* (soluble) that we tested for their activity in NOD2-expressing HEK cells (Fig. 2e and Extended Data Fig. 5). At increasing concentrations of the PGN in WT cells, we observed a robust increase in both IL-8 cytokine production as well as the NF- κ B-reporter assay. The NAGK-deficient cells were unable to respond to either PGN. Taken together, these stimuli represent a more complex physiological ligand for NOD2 stimulation that should encompass all possible breakdown products that arise within the cell. Importantly, these complex PGN preparations

also require NAGK to exert NOD2 stimulatory activity. As such, one can conclude from these data that NAGK is generally required for NOD2 stimulation (and not only for MDP).

In general, it is well-established that NOD2 is a critical PRR for PGN recognition. Needless to say, there is redundancy with other PRR pathways (most importantly NOD1) when immune cells are challenged with pathogens. We are currently unaware of any other defined PGN degradation product that we could test for NAGK and/or NOD2 activity, let alone that we already observe a complete loss of activity when we stimulate NAGK-deficient cells with PGN preparations. Further, in light of the fact that we observe a complete loss in NOD2 stimulatory activity in NAGK-deficient cells treated with either MDP, L18-MDP, MTP or PGN preparations, we don't think that NAGK serves the function to play a modulatory role in the NOD2 pathway, but that it rather constitutes a critical prerequisite for NOD2 activation.

Additional questions:

-Could the authors address why the SLC15A peptide transporters were not identified (redundancy or the use of L18-MDP?).

We assume that the lipophilicity of L18-MDP renders MDP cell permeable, independently of SLC function. Further, given the redundant activity of SLC15A2 and SLC15A4 (PMID: 20406817, 29784761), the gene-trap technology would not have been able to identify these genes. As other forward genetic screening approaches, this approach is unfortunately limited in uncovering genes with redundant functions.

-The authors show that NAGK can phosphorylate MDP in vitro but not that this occurs in cells.

This is a very important point that was missing in the previous manuscript. We employed sensitive mass spectrometry (metabolomics pipeline) on L18-MDP treated WT and NAGK-deficient KBM-7 cells. We successfully detected L18-MDP within treated cells and also observed the conversion of L18-MDP to MDP in both WT and NAGK-deficient cells. The presence of p-MDP was completely dependent NAGK, supporting the essential role of NAGK *in cellulo* (Fig. 3e, f).

-What are all the other significant hits in the screen and will the authors provide all sequencing data and analysis scripts?

The raw data of the gene trapping screen are now appended to this manuscript as Supplementary Table 1. We believe this will be an invaluable resource for the NOD2 field. Moreover, we will provide the raw data of the sequencing results in the final version of this manuscript.

Reviewer Reports on the First Revision:

Referees' comments:

Referee #1 (Remarks to the Author):

The revised manuscript by Stafford et al. is significantly improved and, in my opinion, they have elegantly provided responses to the points raised by the reviewers, including mine. The addition of data with Nagk^{-/-} mice, as well as additional biochemical assays, strengthen their conclusions.

One remaining issue is the new data presented in Fig. 2g. In these data, the authors identified numerous proteins whose phosphorylation status is changed after L18-MDP stimulation, in a NAGK-dependent manner. Because NAGK is itself a kinase, the results, as they are currently presented, give the false impression that NAGK may be directly responsible for the phosphorylation of these numerous proteins, following MDP stimulation. This seems in my opinion very unlikely, as it would argue that there are hundreds of protein targets downstream of MDP that are regulated by NAGK independently of NOD2. It is much more likely that those phospho-proteins are actually simply downstream of NOD2 signaling and are not direct NAGK targets. I assume this is what the authors meant, but it is not clear from the manuscript.

So, my suggestions are the following: either you change the text to prevent this misinterpretation or, if those phosphorylation events are really NAGK-dependent yet NOD2-independent, you provide data showing similar heatmap profile in NOD2 KO cells.

Stephen E. Girardin

Referee #2 (Remarks to the Author):

In this revised manuscript Stafford et al. have sought to address some questions raised in review of their study on the requirement for N-acetylglucosamine kinase (NAGK) to phosphorylate MDP in making MDP (now 6-O-phospho-MDP) an activator of NOD2. The manuscript has been substantially modified as a result. Key strengths of the revision include the generation of NAGK knockout mice, inclusion of data with bone marrow derived macrophages from these mice, and detection of phosphorylated MDP by mass spec in cells. Overall, these are strong additions to the study that make it more robust, and I am firmly convinced that authors have successfully discovered and tested an important new addition to our model of how NOD2 detects MDP.

Some comments on the revision:

With a NAGK knockout mouse in hand, it is somewhat disappointing that the best the authors can do with it is to report that MDP sensing is abrogated in bone marrow derived macrophages. I certainly appreciate the move to evaluating primary cells and "real" inflammatory outputs like cytokine production, and these additions to the study are VERY STRONG. I am still left wondering whether NAGK is really important for natural detection of bacterial cell wall peptidoglycan in host defense. To be fair, I feel that defining a robust non-redundant role for NOD2 in host defense against bacteria

has been a strangely enduring challenge in the NOD2 field. The authors clearly show that NAGK is necessary for recognition of peptidoglycan by NOD2-transfected HEK cells (although how intact peptidoglycan gets into the cytosol of HEK cells is unclear), and from a reductionist point of view, this is nice.

Regarding binding of phospho-MDP to NOD2 I commend the authors' diligent efforts in trying to measure binding affinities. As the authors note, this has been previously attempted with varied and little success. But the present study suggests that MDP may never have been the "optimal" ligand in this work. Unfortunately, use of phospho-MDP doesn't really change the outcomes of this line of investigation. This may well be, as the authors hypothesize, because NOD2 has to be in a "yet-to-be-defined "functional state"" to make these binding assays work. Nevertheless, we are formally stuck not sure that phospho-MDP is a ligand for NOD2 any better than we previously "knew" that MDP is a ligand for NOD2. We do know now that phospho-MDP formation is strongly beneficial in activating NOD2, and that's a substantial finding. The authors should discuss that it is possible that additional MDP modifications could be necessary still for its ultimate proposed interactions with NOD2 (we don't know that further modifications aren't required any better than we previously knew this with regards to unmodified MDP).

Finally, NOD2 is certainly clinically important, and I appreciate its status as a textbook featured PRR. Where and how it is actually critically involved in innate sensing of bacterial peptidoglycan has been strangely hard to establish; it seems like for every paper identifying a non-redundant role there's another contradicting it. If this weren't the case, the authors would feel that it is important to show that the NAGK knockout mice/cells phenocopy the "textbook" biological effect of NOD2 in host defense. It's not reasonable to expect the authors to sort this out, and the ultimate biological roles of NOD2 are not the focus of this study. In any case, the "textbook" model that NOD2 is involved in cytosolic detection of bacterial peptidoglycan degradation products is unchanged. We've added further processing of MDP beyond degradation-based release of soluble fragments, and while not a "paradigm shift", it is certainly necessary now to add to the model.

Referee #3 (Remarks to the Author):

In their revised manuscript entitled 'NOD2 is a sensor for NAGK-phosphorylated muramyl dipeptide' Stafford et al describe a new and unexpected role for NAGK in MDP sensing by NOD2. This manuscript aims to deliver a single message: instead of sensing MDP, NOD2 senses phospho-MDP generated by NAGK.

In their revision quite a few issues that were raised have been addressed, such as the demonstration that indeed MDP becomes phosphorylated in cells in a NAGK-dependent manner. Furthermore, using permeabilized cells the authors address the separate consequences of MDP phosphorylation on uptake vs signaling. Finally, the authors additionally demonstrate that whole peptidoglycan preparations from bacteria activate the pathway in a NAGK-dependent manner. These additions are essential to support their claims.

Nevertheless, for a manuscript that conveys a single message to be published in Nature one would expect a solid support for the claims that are made as well as evidence for the relevance. Therefore, the absence of binding data to support that NOD2 specifically binds to phospho-MDP is problematic. At this stage it remains unclear what NOD2 binds to, it could even be a further modification of phospho-MDP. Furthermore, not showing the relevance of NAGK during actual bacterial infection is also disappointing.

One additional comment; the gaps above - while significant - are not at all reflected in the claims made by the authors. This is most evident in the title: "NOD2 is a sensor for NAGK-phosphorylated muramyl dipeptide". Without binding data, this is a clear overstatement. A title more reflective of the actual data could read "Phosphorylation of muramyl dipeptide by NAGK is required for NOD2 stimulation". Additionally, the Abstract reads "Indeed, mice deficient in NAGK display a complete defect in sensing MDP". This is misleading because only an experiment with isolated cells is presented.

Author Rebuttals to First Revision:

Once again, we would like to thank the reviewers for their valuable time assessing our manuscript in great detail. We are glad to learn that our manuscript is found to be “very strong” and that we “successfully discovered and tested an important new addition to our model of how NOD2 detects MDP”.

Referees' comments:

Referee #1 (Remarks to the Author):

The revised manuscript by Stafford et al. is significantly improved and, in my opinion, they have elegantly provided responses to the points raised by the reviewers, including mine. The addition of data with NAGK^{-/-} mice, as well as additional biochemical assays, strengthen their conclusions.

One remaining issue is the new data presented in Fig. 2g. In these data, the authors identified numerous proteins whose phosphorylation status is changed after L18-MDP stimulation, in a NAGK-dependent manner. Because NAGK is itself a kinase, the results, as they are currently presented, give the false impression that NAGK may be directly responsible for the phosphorylation of these numerous proteins, following MDP stimulation. This seems in my opinion very unlikely, as it would argue that there are hundreds of protein targets downstream of MDP that are regulated by NAGK independently of NOD2. It is much more likely that those phospho-proteins are actually simply downstream of NOD2 signaling and are not direct NAGK targets. I assume this is what the authors meant, but it is not clear from the manuscript.

So, my suggestions are the following: either you change the text to prevent this misinterpretation or, if those phosphorylation events are really NAGK-dependent yet NOD2-independent, you provide data showing similar heatmap profile in NOD2 KO cells.

Stephen E. Girardin

We kindly thank you for your constructive comments. We really appreciate the positive feedback from one of the pioneers of the NOD field for our work.

To address your concern regarding the presentation of the phosphoproteome data, we concur that an additional sentence really helps to clarify the role of NAGK in this context. As such, we added the following sentence when presenting these data: “These results are consistent with the notion that NAGK-deficiency completely blunts MDP-dependent signal transduction.”

Referee #2 (Remarks to the Author):

In this revised manuscript Stafford et al. have sought to address some questions raised in review of their study on the requirement for N-acetylglucosamine kinase (NAGK) to phosphorylate MDP in making MDP (now 6-O-phospho-MDP) an activator of NOD2. The manuscript has been substantially modified as a result. Key strengths of the revision include the generation of NAGK knockout mice, inclusion of data with bone marrow derived macrophages from these mice, and detection of phosphorylated MDP by mass spec in cells. Overall, these are strong additions to the study that make it more robust, and I am firmly convinced that authors have successfully discovered and tested an important new addition to our model of how NOD2 detects MDP.

Some comments on the revision:

With a NAGK knockout mouse in hand, it is somewhat disappointing that the best the authors can do with it is to report that MDP sensing is abrogated in bone marrow derived macrophages. I certainly appreciate the move to evaluating primary cells and “real” inflammatory outputs like cytokine production, and these additions to the study are VERY STRONG. I am still left wondering whether NAGK is really important for natural detection of bacterial cell wall peptidoglycan in host defense. To

be fair, I feel that defining a robust non-redundant role for NOD2 in host defense against bacteria has been a strangely enduring challenge in the NOD2 field. The authors clearly show that NAGK is necessary for recognition of peptidoglycan by NOD2-transfected HEK cells (although how intact peptidoglycan gets into the cytosol of HEK cells is unclear), and from a reductionist point of view, this is nice.

Regarding binding of phospho-MDP to NOD2 I commend the authors' diligent efforts in trying to measure binding affinities. As the authors note, this has been previously attempted with varied and little success. But the present study suggests that MDP may never have been the "optimal" ligand in this work. Unfortunately, use of phospho-MDP doesn't really change the outcomes of this line of investigation. This may well be, as the authors hypothesize, because NOD2 has to be in a "yet-to-be-defined "functional state"" to make these binding assays work. Nevertheless, we are formally stuck not sure that phospho-MDP is a ligand for NOD2 any better than we previously "knew" that MDP is a ligand for NOD2. We do know now that phospho-MDP formation is strongly beneficial in activating NOD2, and that's a substantial finding. The authors should discuss that it is possible that additional MDP modifications could be necessary still for its ultimate proposed interactions with NOD2 (we don't know that further modifications aren't required any better than we previously knew this with regards to unmodified MDP).

We kindly thank the reviewer for their continued positive and constructive feedback and we are really happy this reviewer now finds our study to be "very strong" and an important step forward for the field. We concur with the point raised by the reviewer that additional, yet-to-be-defined steps could be required for NOD2 activation. We updated the discussion accordingly and now mention that without structural insight, there could be still room for additional steps downstream of MDP phosphorylation.

As such, we now write: "However, in the absence of biochemical data demonstrating binding, this remains speculative, and it is conceivable that other intermediate steps are required for NOD2 to recognise phosphorylated muramyl peptides."

Finally, NOD2 is certainly clinically important, and I appreciate its status as a textbook featured PRR. Where and how it is actually critically involved in innate sensing of bacterial peptidoglycan has been strangely hard to establish; it seems like for every paper identifying a non-redundant role there's another contradicting it. If this weren't the case, the authors would feel that it is important to show that the NAGK knockout mice/cells phenocopy the "textbook" biological effect of NOD2 in host defense. It's not reasonable to expect the authors to sort this out, and the ultimate biological roles of NOD2 are not the focus of this study. In any case, the "textbook" model that NOD2 is involved in cytosolic detection of bacterial peptidoglycan degradation products is unchanged. We've added further processing of MDP beyond degradation-based release of soluble fragments, and while not a "paradigm shift", it is certainly necessary now to add to the model.

We agree with the reviewer that sorting out the role of NAGK in vivo would clearly go beyond the scope of this study, in which we describe a new component of the PGN recognition pathway.

Referee #3 (Remarks to the Author):

In their revised manuscript entitled 'NOD2 is a sensor for NAGK-phosphorylated muramyl dipeptide' Stafford et al describe a new and unexpected role for NAGK in MDP sensing by NOD2. This manuscript aims to deliver a single message: instead of sensing MDP, NOD2 senses phospho-MDP generated by NAGK.

In their revision quite a few issues that were raised have been addressed, such as the demonstration that indeed MDP becomes phosphorylated in cells in a NAGK-dependent manner. Furthermore, using permeabilized cells the authors address the separate consequences of MDP phosphorylation on uptake vs signaling. Finally, the authors additionally demonstrate that whole peptidoglycan preparations from bacteria activate the pathway in a NAGK-dependent manner. These additions are essential to support their claims.

Nevertheless, for a manuscript that conveys a single message to be published in Nature one would expect a solid support for the claims that are made as well as evidence for the relevance. Therefore, the absence of binding data to support that NOD2 specifically binds to phospho-MDP is problematic. At this stage it remains unclear what NOD2 binds to, it could even be a further modification of phospho-MDP. Furthermore, not showing the relevance of NAGK during actual bacterial infection is also disappointing.

We appreciate the reviewer's acknowledgment of our efforts to obtain binding studies of 6-phospho-MDP with NOD2. However, we would kindly note that the NOD receptors have been exceptionally resistant to researchers' attempts to obtain biochemical insights regarding their direct ligand recognition. We believe that our work will rekindle interest in this endeavor and lay the foundation of a structure of ligand-bound NOD2. We hope that the reviewer agrees that it would be beyond the scope of this initial discovery to solve this longstanding problem.

Nevertheless, not to overstate our findings—and in agreement with the reviewer's comment regarding that matter—we now additionally emphasise in the discussion that additional factors downstream of NAGK-dependent MDP phosphorylation might be required for NOD2 activation.

Please refer to the following sentence in the discussion: "However, in the absence of biochemical data demonstrating binding, this remains speculative, and it is conceivable that other intermediate steps are required for NOD2 to recognise phosphorylated muramyl peptides."

One additional comment; the gaps above - while significant - are not at all reflected in the claims made by the authors. This is most evident in the title: "NOD2 is a sensor for NAGK-phosphorylated muramyl dipeptide". Without binding data, this is a clear overstatement. A title more reflective of the actual data could read "Phosphorylation of muramyl dipeptide by NAGK is required for NOD2 stimulation". Additionally, the Abstract reads "Indeed, mice deficient in NAGK display a complete defect in sensing MDP". This is misleading because only an experiment with isolated cells is presented.

We completely agree with this reviewer's comment and changed both the title and the abstract according to these suggestions.